# Cyborg insect factory: automatic assembly for insect-computer hybrid robot via vision-guided robotic arm manipulation of custom bipolar electrodes

Qifeng Lin, Nghia Vuong, Kewei Song, Phuoc Thanh Tran-Ngoc, Greg Angelo Gonzales Nonato & Hirotaka Sato ✉

Insect–computer hybrid robots offer strong potential for navigating complex terrains. This study identified the intersegmental membrane between the pronotum and mesothorax of the Madagascar hissing cockroach as an effective site for electrical stimulation to control direction and speed. A pair of bipolar electrodes was custom-designed, and an automatic assembly system was developed, integrating a robotic arm, vision-based site detection, and an insect fixation structure. The system achieved assembly in 68 s. Hybrid robots exhibited robust steering (over 70°) and deceleration (68.2% speed reduction) with performance comparable to manually assembled counterparts. Controlled navigation along an S-shaped path confirmed accurate directional control. Furthermore, a multi-agent system of four hybrid robots covered 80.25% of an obstructed terrain in 10 minutes and 31 seconds. This work demonstrates a scalable strategy for automating the fabrication of insect–computer hybrid robots, enabling efficient and reproducible assembly process while maintaining effective locomotion control.

Various insect-scale robots have been engineered, demonstrating exceptional maneuverability within complex and narrow terrains[1–5]. This capability has spurred advancements in mechanically structured and insect-computer hybrid robots (i.e., biobots or cyborg insects). While these robots have a similar size, they possess unique self-locomotion energy sources[3,6,7] and adaptability to challenging terrains[8]. Therefore, such robots' potential is increasingly explored as a robotic platform[3,9–13] across various applications.

To achieve locomotion control in insects, researchers have studied stimulation electrodes targeting their muscles, neuron systems, and sensory organs[7,14–17]. Both invasive[2,3,15] and non-invasive electrodes[7] have been manually implanted into the insects' target body parts to enhance stimulation. However, the intricate and fragile anatomy of insects makes the manual surgery process time-consuming (~15 min per insect[18]) and difficult[15]. For instance, insects' small and soft

antennae and cerci necessitate the use of specialized microinstruments and microscopes to manipulate their tissues accurately. Even minimal force can cause unintended shape changing, making procedures highly demanding (Supplementary Fig. S1). Furthermore, the success of the surgical procedure significantly reliant on the operator's expertise[2,3,15], which impacts the risk of accidental injury to the insects. Due to variations in the operator's surgical skills, even with the same implantation method, the insect–computer hybrid robots operated by different operators may behave differently.

For consistent production of insect-computer hybrid robots, transitioning from manual to automatic assembly processes is imperative, particularly for applications that demand large scale deployment, such as post-disaster search and rescue or factory inspections, where multiple robotic agents significantly enhance efficiency than a single unit[19,20]. Therefore, developing automatic assembly methods for insect-

School of Mechanical & Aerospace Engineering, Nanyang Technological University, Singapore, Singapore. ✉e-mail: hirosato@ntu.edu.sg

computer hybrid robots is crucial for mass production. Madagascar hissing cockroach has been used in various applications as a powerful platform[3,8,13,21]. Consequently, our study focuses on optimizing its mass production. Although these insects generally share a similar body structure, individual size variations pose challenges to achieving uniform localization of the implantation site in contrast to some assembly tasks for mechanical parts of certain shapes[22]. Consequently, to ensure precise localization of the implantation site, advanced deep learning techniques are employed.

To control the locomotion direction of Madagascar hissing cockroach, stimulation protocols have been developed on the insects' antennae and abdomen[3,15,23]. The antennae detect and navigate around the obstacles[24], and the tactile stimuli[25] significantly influence cockroach movement. The electrical stimulation of the antennae effectively induces directional turning among the insects[7,15]. However, the antennae are soft, fragile, and tiny[15] (with a 0.6–0.7 mm diameter, Fig. 1B), posing challenges in securely attaching and implanting electrodes. Moreover, antennae used in hybrid robots to enable autonomous obstacles navigation[7,26] destroy the insects' innate ability to maneuver around the obstacles[7,27]. Therefore, this study excludes antennae as the target site for stimulation.

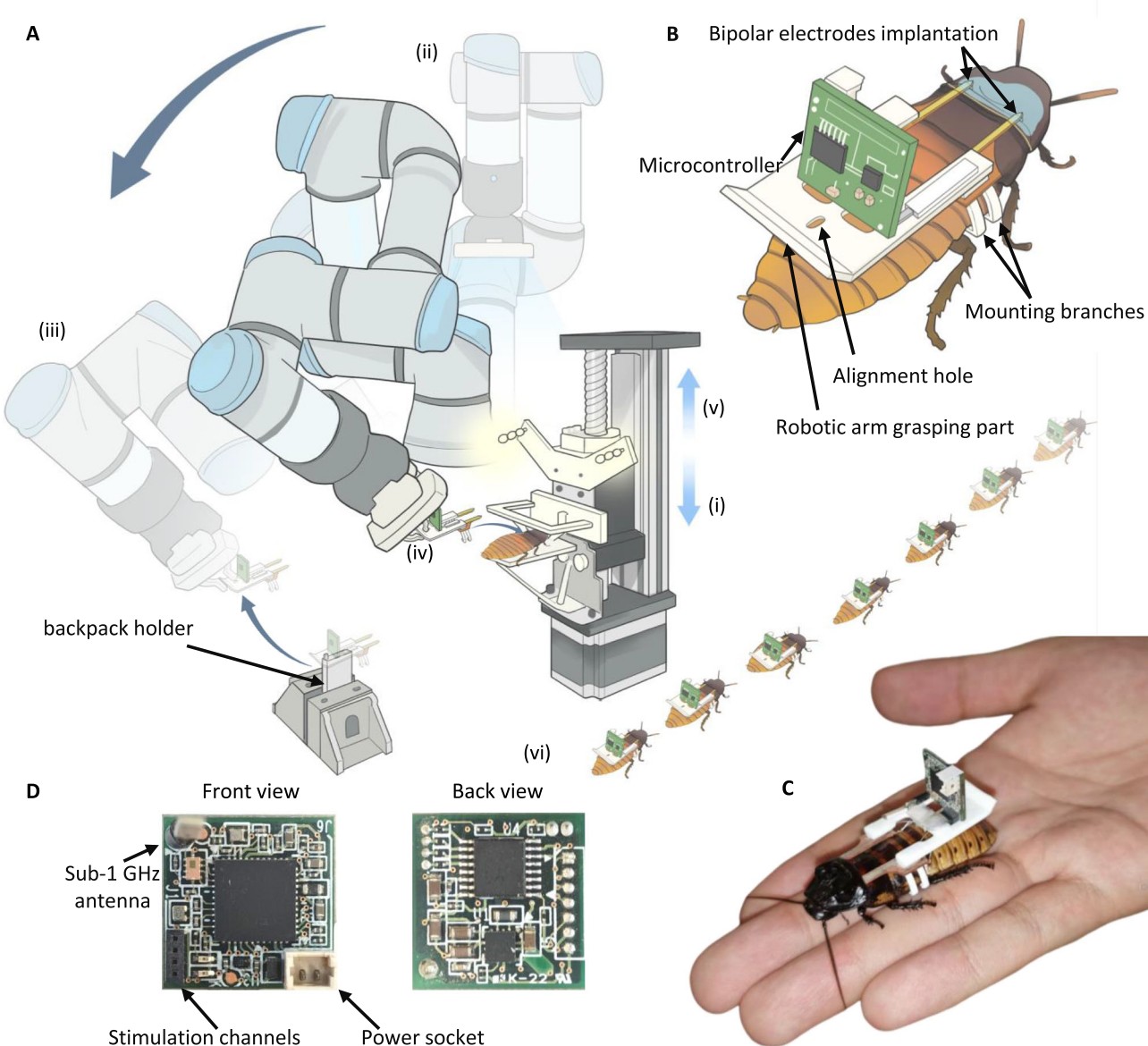

**Fig. 1 | Insect-computer hybrid robot and its automated assembly configuration. A** An anesthetized cockroach secured for backpack assembly using the robotic arm. The automatic assembly included an insect fixation structure driven by a slide motor, a robotic arm equipped with a gripper for grasping and assembling the backpack, and a depth camera for precise localization of the insect's body position during assembly. **B** Components of insect-computer hybrid robot. The white mounting structure was attached with a microcontroller and bipolar electrodes. The grasping section of the backpack was held by a robotic arm gripper, while the alignment hole placed the backpack. Mounting branches were used to hook the insect's metathorax. Bipolar electrodes of the backpack were implanted into the intersegmental membrane between the insect's pronotum and mesothorax. **C** Insect-computer hybrid robot. The backpack-assembled insect was controlled to execute turning maneuvers and decelerating. **D** A microcontroller with Sub-1GHz communication used to between the insect-computer hybrid robot and workstation. Stimulation signals were transmitted from the stimulation channels to the bipolar electrodes for insects' locomotion control. After the assembly finished and the hybrid robot was needed for the locomotion control, a LiPo battery was plugged inside the power socket. Double-sided tapes were used to stick the battery to the backpack.

Stimulating the abdomen's sides influences the locomotion direction of the insects[3]. However, cockroaches' abdominal cuticles are short and thin—with the third abdominal cuticle measuring 3.8–5.0 mm in length and 0.2–0.3 mm in thickness—making automatic electrode implantation difficult. Consequently, an alternative stimulation site should be explored. Building upon studies on similar terrestrial platforms such as Zophobas morio, which alters its locomotion direction when stimulated on the pronotum and elytra[2], we hypothesized that pronotal stimulation would similarly alter the cockroach's direction. Notably, the pronotum cuticle is both larger and thicker than the abdominal cuticles, measuring ~11.6–13.4 mm in length, 0.5–0.6 mm in thickness, which eases attachment and detection. Additionally, akin to the interspace between abdominal segments[7], an intersegmental membrane between the pronotum and the mesothorax is the key target site for stimulation, enabling directional control of the insect (Fig. 1C).

This study proposes a stimulation protocol designed to control an insect-computer hybrid robot. To analyze the insects' responses to electrical stimulation, we recorded their neural activities, foreleg movements, and locomotion patterns during simulation. A backpack was developed, integrating microcontroller, stimulation electrodes, and a mounting device (Fig. 1B, C). The hybrid robot was controlled wirelessly for steering and stopping. Next, an automatic assembly was developed, incorporating a slide motor, a fixation structure for the insect, an intel RealSense D435 camera, a Robotiq Hand-e gripper and a Universal Robot UR3e (Fig. 1A). This system relied on the visual detection of the target body position. During the assembly process, the insect was fixed in a structure mounted on a motor-driven slider. After assembling the backpack into the insect, the insect fixation structure was released, completing the assembly of the hybrid robot. Five automatically assembled hybrid robots were studied for locomotion control, including steering, and deceleration, with their performance compared to that of the manually assembled ones. Four hybrid robots traversed an outdoor uneven terrain using a UWB localization system.

## Results and dicussion

### Bipolar electrode

Due to the challenges of securing the antennae and abdomen and implanting electrodes with the robotic arm, this study focuses solely on the pronotum. To facilitate turning locomotion in cockroaches and minimize the number of implantation sites, a pair of bipolar electrodes was designed and implanted on the left and right sides of the intersegmental membrane between the pronotum and mesothorax (Figs. 1C and 2A). Each bipolar electrode comprised a copper pattern for electrical signal transmission and a microneedle structure to rapidly puncture the intersegmental membrane, and a hook mechanism to prevent detachment after implantation.

The complex structural features and the combination of plastic and metal components in bipolar electrodes necessitated specialized fabrication processes. Integrating multi-material 3D printing technology with an electroless plating process offered an effective approach for fabricating 3D electronic structures with spatial configuration and electrical signal carrying functions[28–30] (Fig. 2B, Supplementary Movie S1). Initially, multi-material DLP3D printing was employed to fabricate the precursor structure of the bipolar electrodes, integrating a normal resin with an active precursor. The active precursor contains a catalytic agent that facilitates selective metal deposition during electroless plating, allowing metallization on both sides of the structure. Figure 2C presents the precursor and the final bipolar electrode with selectively deposited copper.

To ensure effective implantation and electrical stimulation, the bipolar electrodes should possess high hardness and toughness. Therefore, ABS-like photosensitive resins[31] was chosen as the material for electrode fabrication (in their normal resin form or as an active precursor). Finite element simulations (Fig. 2D, i–iv) showcased bipolar electrodes exhibiting a uniform stress distribution and controlled deformation during implantation, without experiencing damage or yielding. Additionally, the designed bipolar electrodes were securely implanted by monitoring the microtip's stabbing stress at the membrane (Fig. 2D, v).

To address the delamination or separation of metal plating during implantation, we calibrated the plating adhesion by the ASTM D3359-09 standard. We incorporated chemical etching into the selective electroless plating to enhance the adhesion and achieve a high 4B grade (Fig. 2E). Additionally, the implanted part comprising cockroach's soft tissue (intersegmental membrane) and electrolyte solution, exerted minimal cutting force on the electrode, reducing impact (Fig. 2A). The impedance profile (Fig. 2F) of a plated layer on the bipolar electrode reveals that the impedance remained consistently below $70\,\Omega$, considerably lower than the impedance of comparative non-invasive electrodes (exceeding $1000\,\Omega$[7]). This reduced impedance facilitated stronger stimulation and a more pronounced insect reaction.

The electrical conductivity and the thickness of the plating were $3.12 \times 10^7$ S/m and $2.5\,\mu m$, respectively. The copper-replacing-nickel plating method caused the electrodes to achieve selective copper metallization. After 5 min of immersion in the plating solution, a substantial increase in plating thickness was observed (Fig. 2G). A thicker plating layer enhanced conductivity, reduced impedance, and improved corrosion resistance; however, it increased parasitic capacitance. This study selected a plating time of 16 min for plating thickness and conductivity optimization. By controlling the electroless plating's duration, the plated layer's thickness on the bipolar electrodes was precisely modulated, enabling the fine-tuning of conductivity and other electrochemical properties.

### Stimulation protocol on insect's pronotum

The width of the intersegmental membrane between pronotum and mesothorax was measured ($1.4 \pm 0.2$ cm) across cockroaches with different sizes. To ensure that the bipolar electrodes implanted into the membrane, the distance between the two bipolar electrodes was set as 1.0 cm to satisfy minimum width of the membrane, i.e., 1.2 cm.

To optimize stimulation voltage, neural activities in the insects' neck region were recorded and analyzed (Fig. 3A). A progressive increase in the number of detected neural spikes was observed as the voltage increased from 0.5 V to 3.0 V, exhibiting heightened sensitivity to stronger electrical stimulation. A plateau at 3.0–3.5 V indicated that electrical stimulation above 3.0 V did not produce a strong neural response from the insects. However, increasing the stimulation voltage to 4.0 V yielded a 23.5% decline in the average number of spikes, denoting reduced neural activity due to potential damage to the insects' neural system[32–34]. To prevent unnecessary damage and maintain effective stimulation, a 3.0 V voltage was considered optimal for the subsequent discussions.

Since the pronotum of the insect is connected to its forelegs (Fig. 3B) which guides its locomotion[35], examining the forelegs' status during stimulation is crucial. When one side of the pronotum was stimulated, the foreleg on that side contracted until the electrical stimulation was discontinued (Fig. 3B, Supplementary Movie S2), confirming that the stimulation directly influenced forelegs. Given that the forelegs guide the insects during walking[35], this observation suggests their usage in controlling the insect's orientation. Consequently, this study explored locomotion in detail.

To simplify future automatic assembly processes, the stimulation bipolar electrodes, microcontroller, and mounting parts were integrated into a backpack. These backpacks were manually and then automatically affixed to insects to compare locomotion control. The manually assembled hybrid robots were tested for their locomotion control ($N = 5$ insects). The insects' responses to the electrical

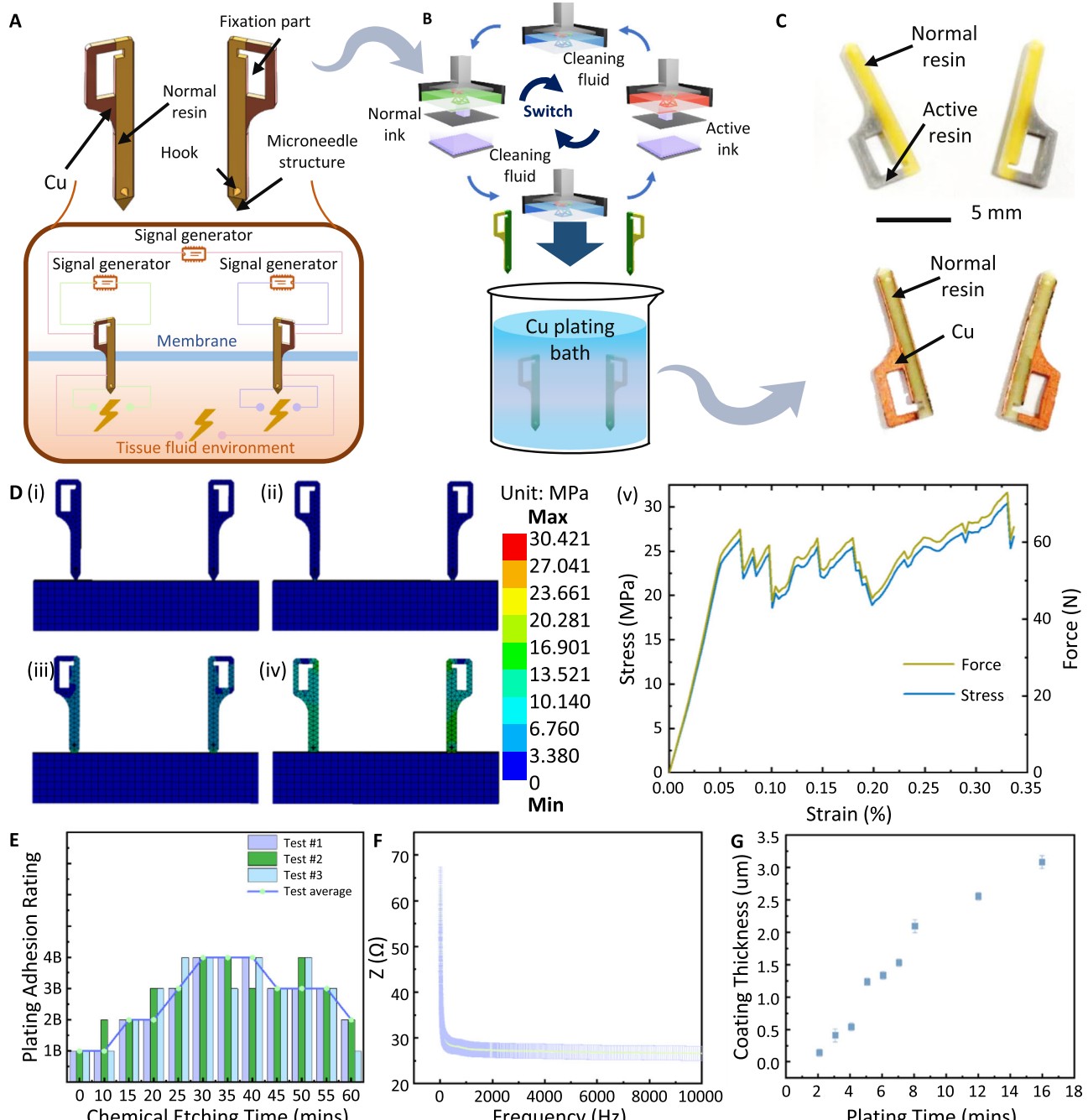

**Fig. 2 | Design, fabrication, and properties of the custom bipolar electrode.**
**A** Custom-designed bipolar electrode utilized a microneedle structure integrated with a hook design, facilitating rapid membrane penetration and secure self-locking within the punctured membrane. The electrode comprised normal resin and patterned copper wires to transmit stimulation. **B** Bipolar electrodes fabricated using multi-material 3D printing technology followed by electroless plating. **C** Bipolar electrodes before and after copper plating. **D** Finite element modeling and analysis during the process of bipolar electrodes implantation into the intersegmental membrane between the cockroach's pronotum and mesothorax. **E** Adhesion rating of the metal plating within the bipolar electrode, evaluated using the ASTM D3359-09 standard. Incorporating chemical etching strengthened the metal adhesion, preventing detachment during implantation and use and preserving electrical performance. **F** Impedance of conductive bipolar electrodes with conductivity (<70 Ω), significantly lower than non-invasive electrodes' impedance[7]. **G** Plating thickness vs. time. The curve depicts a growth trend, representing a thicker coating layer with a longer plating duration.

stimulation applied to both sides of pronotum (Fig. 3C) confirmed that the implanted stimulation electrodes, positioned within the intersegmental membrane between pronotum and mesothorax, induced directional movement in the insects. Electrical stimulation turn the insects to the left and right with average angles of 68.0° and 82.6°, and maximum angular speeds of 275.8°/s and 298.2°/s, respectively. Additionally, the simultaneous contraction of the forelegs during stimulation (Fig. 3B) suggests that the implantation of the insect's

intersegmental membrane between the pronotum and mesothorax successfully stimulated the forelegs, with the insect steered with unilateral stimulation of a foreleg.

Our stimulation protocol outperforms previous non-invasive electrode-based methods[7]. It enhances the maximum steering speed by over five times and increases the turning angle by over 76.6%. Additionally, this protocol requires only 40% of the stimulation time and 75% of the stimulation voltage. These enhancements indicate that

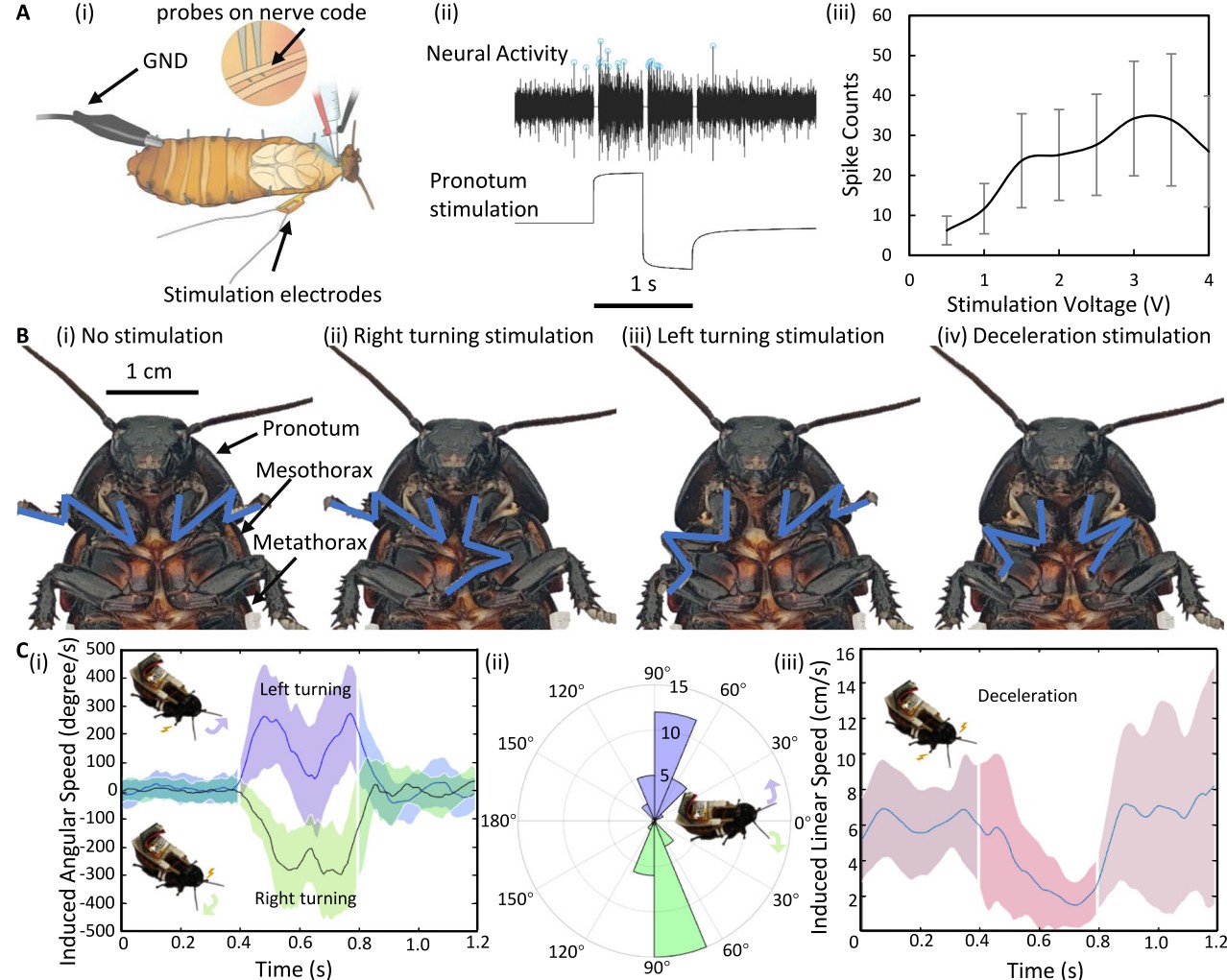

**Fig. 3 | Analysis of insect's neural responses to electrical stimulation. A** Insect's neural activity in response to electrical stimulation. i Neural recording configuration. Two probes were fixed to the nerve cord within the cockroach's neck to record neural signals. ii Neural activity and corresponding electrical stimulation. Neural spikes induced by electrical stimulation (blue circles) were quantified to assess responses across various stimulation voltages. iii Spike counts at varying stimulation voltages (mean ± SD). For each voltage, nine trials of recording were conducted. **B** Insect leg reactions to electrical stimulation. i No stimulation: The insect's forelegs remained extended. ii Right-turning stimulation: The insect's left foreleg was stimulated and contracted. iii Left-turning stimulation: The insect's right foreleg was stimulated and contracted. iv Deceleration stimulation: Both insect's forelegs were stimulated and contracted. **C** Responses of insect locomotion to electrical stimulation. i Induced angular speed during turning stimulation (mean ± SD). ii Angular variations during turning stimulation (mean ± SD. iii Induced linear speed during deceleration stimulation (mean ± SD).

our protocol triggered more intense turning responses with reduced time and energy consumption, signifying a more resource-efficient approach, improving the hybrid robot's overall performance, operational efficiency, and cost-effectiveness. Our approach optimizes locomotion control for insect-computer hybrid robots, making it valuable for practical applications that demand quick and energy-efficient manoeuvrability.

Furthermore, the cockroach decelerated when the electrical stimulation was outputted from the outer electrodes of the bipolar electrode pair (Fig. 2A), making the first successful implementation of the deceleration control in insect-computer hybrid robots. For the control of the insect-computer hybrid robot, both direction control and speed control are fundamental. Previously designed direction control, including left and right turning speed realized the full direction control of the insect-computer hybrid robot. However, for the speed control, only acceleration stimulation was realized. Hence, our discovery on the deceleration control could realize the speed control more fully.

After 0.33 s of stimulation, the insects' average walking speed declined from 6.2 cm/s to a minimum of 1.5 cm/s (Fig. 3C, iii),

translating their body length/s to reduce from 112.9% to 27.6% with an average body length of 5.5 cm. Consequently, the insects experienced an 85.3% deceleration relative to their body length. The standard deviation of the minimum speed normalized with the mean speed during stimulation was 0.83. This small normalized standard deviation in the minimum decelerated speed highlights the consistency of the deceleration stimulation. The simultaneous contraction of both forelegs during stimulation (Fig. 3B, iv) further highlights the direct correlation of the deceleration trend in cockroaches with the stimulation of their forelegs.

Our findings prove that the proposed stimulation protocol on the pronotum realized steering and deceleration control of the insect-computer hybrid robot (Supplementary Movie S3).

### Vision-based automatic assembly of insect-computer hybrid robots

The automatic assembly of hybrid robots comprises the following steps: 1) The pronotum and mesothorax of an anesthetized insect are secured and the intersegmental membrane is exposed. 2) A reference

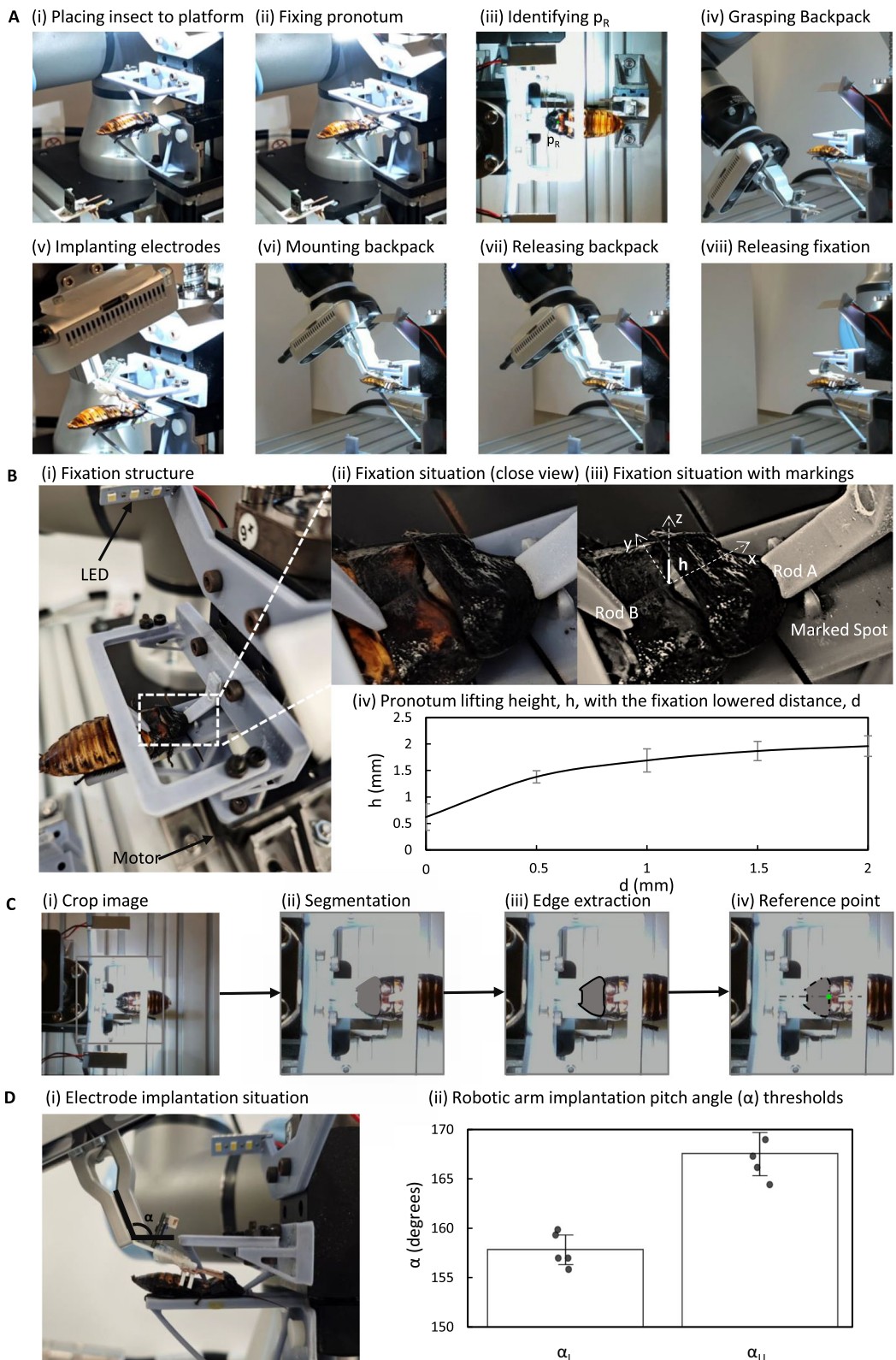

**A** (i) Placing insect to platform  (ii) Fixing pronotum  (iii) Identifying $p_R$  (iv) Grasping Backpack

(v) Implanting electrodes  (vi) Mounting backpack  (vii) Releasing backpack  (viii) Releasing fixation

**B** (i) Fixation structure  (ii) Fixation situation (close view)  (iii) Fixation situation with markings

(iv) Pronotum lifting height, h, with the fixation lowered distance, d

**C** (i) Crop image  (ii) Segmentation  (iii) Edge extraction  (iv) Reference point

**D** (i) Electrode implantation situation  (ii) Robotic arm implantation pitch angle (α) thresholds

point is identified for electrode implantation. 3) The backpack is grasped using a robotic arm's gripper. 4) The bipolar electrodes are implanted into the exposed membrane utilizing the robotic arm. 5) Force is applied to the backpack until its mounting branches latch onto the insect's metathorax. 6) The backpack is disengaged from the gripper. 7) The robotic structure is retracted to release the insect (Fig. 4A).

Key considerations that must be addressed for successful assembly are as follows. First, the insect's pronotum and mesothorax must be fixed to reveal their intersegmental membrane (Fig. 4A, ii). Second, the reference point on the pronotum must be identified for accurately implanting the bipolar electrodes by the robotic arm. Finally, the robotic arm must be maneuvered to assemble the backpack onto the insect at an optimal angle for secure assembly.

**Fig. 4 | Automated assembly of insect-computer hybrid robots. A** Automated assembly. i, ii An anesthetized insect on the platform was secured with a 3D customized structure. iii Insect along with the detected reference point for bipolar electrode implantation (green dot). iv–vi The robotic arm grasping a backpack, implanting bipolar electrodes, and mounting the backpack. vii, viii The robotic arm releasing the backpack to allow the fixation structure to retract. **B** Insect fixation. i Configuration for insect fixation. ii Magnified view of the insect's pronotum fixation. iii Details of the fixation with markings. Rods A and B pressed the anterior pronotum and the mesothorax, respectively, elevating the posterior pronotum. iv Pronotum lifting height $h$ increased as the distance $d$ of Rod A decreased. For each $d$ value, lifting heights $h$ of ten insects were recorded. The error bars denote the standard deviation. **C** Detection of the implantation reference point. i A square region around the pronotum of the insect cropped for segmentation inference. ii Pronotum of the insect identified using the TransUnet model. iii) Edge of pronotum was identified using the detected pronotum mask. iv Midpoint of the posterior pronotum edge detected as the implantation reference point. **D** Implantation of the bipolar electrode. i Measurement of the implantation pitch angle ($\alpha$) while preventing collisions with the insect or the fixation structure. ii Recording of lower threshold $\alpha_L$ and upper threshold $\alpha_U$ when the backpack touched the 3D-customized structure and the insect, respectively. Five trials of the experiment were conducted. The error bars denote the standard deviation. To minimize the risk of potential collisions, the midpoint of these thresholds, $\alpha = 162.7°$, was selected.

**Exposure of intersegmental membrane between pronotum and mesothorax.** The intersegmental membrane between the pronotum and mesothorax is concealed beneath the hard cuticle of the connected pronotum (Fig. 4B, ii). The posterior pronotum is elevated from the mesothorax to implant bipolar electrodes within this membrane (Fig. 4B, ii) using the developed Rod A and Rod B (Fig. 4B, iii). Rods A and B exerted pressure to the anterior pronotum and the mesothorax, respectively, exposing the membrane to enable electrodes implantation (Fig. 4B, iii).

Initially, Rod A was positioned 4.0 mm above the platform, corresponding to a lowered distance, $d$, of 0 mm. The relationship between the lowered distance, $d$, and the lifting height, $h$, of the pronotum (Fig. 4B, iii) highlights the progressive exposure of the intersegmental membrane as the structure is lowered. The bipolar electrode's thickness of 0.6 mm, required the lifting height, $h$, to consistently exceed this threshold to ensure sufficient space for implantation. On average, when $d \geq 1.5$ mm, the height, $h$, reaches 1.9 mm which provides adequate exposure for successful implantation.

For $d \geq 3.5$ mm, $h$ stabilizes around 1.9 mm, suggesting that further lowering the structure does not significantly increase membrane exposure (Student's $t$ test for $d = 1.5$ and 2.0 mm, and $P = 0.31$). As the lifting heights for both $d = 1.5$ mm and $d = 2.0$ mm surpass twice the thickness of the bipolar electrodes, $d = 1.5$ mm was chosen to prevent excessive pressure on the insect's body while ensuring adequate membrane exposure for bipolar electrode insertion. This approach minimizes the potential injury to the insect, ensuring its physical integrity throughout implantation.

**Detection of pronotum using deep learning.** To implant bipolar electrodes into the exposed intersegmental membrane (Section "Exposure of intersegmental membrane between pronotum and mesothorax"), a computer vision system was employed to determine the membrane's location. Since bipolar electrodes on both sides of the backpack needed simultaneous implantation symmetric to the insect's intersegmental membrane, a reference point $p_R$ was established at the middle point of the posterior pronotum edge. However, the pronotum varied in size and shape across each insect (Supplementary Fig. S5). Although the structure restrained the fixing of the insects' pronotum and mesothorax, the pronotum's position may still vary along the x- and y- axes (Fig. 4B, iii, Supplementary Fig. S5). Therefore, implementing a deep learning-based computer vision model is crucial for accurately identifying the pronotum and determining the location of the reference point, $p_R$.

An evaluation was conducted on several widely used segmentation models, including UNet[36], Deeplabv3[37], TransUNet[38], as well as recently developed segmentation models, such as Segment Anything[39], Segment Anything2[40], LM-Net[41], InceptionNeXt[42], EMCAD[43] and SHViT[44], for their accuracy in pronotum segmentation. The models' performance was measured by the mean intersection over union (mIoU) score, mean Dice similarity coefficient (mDSC), and mean squared error (MSE) of $p_R$ prediction (Table 1).

The TransUNet model surpassed other models based on mIoU, mDSC and MSE metrics due to its hybrid encoder, which leverages the strengths of the Transformer architecture and preserves locality through its convolutional neural network (CNN). The other hybrid models, LM-Net, SHViT and EMCAD were designed to be lightweight models geared towards faster inference times, causing poor performance, apart from LM-Net, which had achieved comparable scores with the other models despite having significantly less parameters. The CNN-based models UNet, Deeplabv3 and InceptionNeXt managed to achieve better segmentation accuracies compared to the foundational ViT-based models SAM and SAM2, with Deeplabv3 even having similar performance to TransUNet in all three metrics as it was generally able to segment out most of the pronotum but did not precisely identify the lower border of the pronotum, causing slightly poorer scores on the three metrics. Subsequently, DSC loss and Boundary Difference Over Union (bDoU) loss[45] were compared with BCE loss to achieve improved boundary results. The results of the loss function evaluation were shown in Table 2 in terms of mIoU score, mDSC score, and MSE of $p_R$ prediction. The DSC and BCE loss functions generally outperformed the bDoU loss function on the mIoU and mDSC metrics, as they evaluated the prediction the full pronotum while the bDoU loss focused primarily on the boundaries of the segmented objects. However, the bDoU loss function achieved the best MSE score among the three, highlighting its effectiveness in training the model to learn object boundaries. Ultimately, the chosen deep learning solution was the TransUNet model trained with the BCE loss function (Fig. 4C), as it had the best mIoU and mDSC scores and only slightly underperformed on the MSE metric compared to the model trained with bDoU loss.

Data ablation study was also conducted to evaluate how the effect of different data augmentation methods on the training images impacts the segmentation performance. Performance based on mIoU, mDSC and MSE of $p_R$ prediction, is shown in Table 3. The TransUNet model had achieved significantly better scores across all three metrics when trained with augmented data, compared to when trained with only original data. The model trained on the asymmetrically scaled augmented data had shown to have slightly better performance in segmenting the boundaries compared to the model trained on the rotated augmented data, attributed to the robustness to variations in the shapes of pronotums. However, the latter had a better segmentation accuracy, as indicated by the larger mIoU and mDSC scores.

**Manipulation of robotic arm.** To assemble the backpack onto the insect, the robotic arm scanned the insect with the camera and identified the reference point for bipolar electrode implantation (Section "Detection of pronotum using deep learning"). The Robotiq Hand-e gripper secured the grasp on the backpack and allowed stable electrode implantation with a gripping force of up to 185 N. Based on the combined weight of the Robotiq Hand-e (1.0 kg), RealSense D435 camera (75.0 g), and the backpack (2.3 g), the Universal Robots UR3e was selected as the robotic arm due to its 3.0 kg payload capacity and a pose repeatability of 0.03 mm at full payload[46]. Additionally, the UR3e's

**Table 1 | Comparison of models on the cockroach test samples dataset**

| Model | Params (M) | mIoU | mDSC | MSE ($p_R$) |
|---|---|---|---|---|
| LM-Net | 5 | 0.8969 | 0.9448 | 1.970 |
| SHViT | 18 | 0.8186 | 0.8990 | 3.722 |
| EMCAD | 26 | 0.7688 | 0.8689 | 3.184 |
| UNet | 31 | 0.8741 | 0.9318 | 2.382 |
| Deeplabv3 | 61 | 0.9245 | 0.9605 | 1.786 |
| TransUNet | 105 | 0.9326 | 0.9650 | 1.749 |
| InceptionNeXt | 193 | 0.8567 | 0.9222 | 2.603 |
| Segment Anything2 | 224 | 0.8040 | 0.8885 | 3.246 |
| Segment Anything | 636 | 0.8471 | 0.9161 | 2.387 |

**Table 2 | Comparison of loss functions used for TransUNet training**

| Loss Function | mIoU | mDSC | MSE ($p_R$) |
|---|---|---|---|
| BCE Loss | 0.9326 | 0.9650 | 1.749 |
| DSC Loss | 0.9289 | 0.9630 | 1.807 |
| bDoU Loss | 0.9283 | 0.9626 | 1.617 |

**Table 3 | Comparison of TransUNet models trained on datasets with various levels of data ablation**

| Training image data | | | Metrics | | |
|---|---|---|---|---|---|
| Original | Asymmetrically scaled | Rotated | mIoU | mDSC | MSE |
| ✓ | | | 0.8721 | 0.9296 | 2.326 |
| ✓ | ✓ | | 0.8847 | 0.9347 | 2.090 |
| ✓ | | ✓ | 0.9211 | 0.9588 | 2.152 |
| ✓ | ✓ | ✓ | 0.9326 | 0.9650 | 1.749 |

500 mm reach supports the RealSense D435's minimum depth sensing distance of ~280.0 mm when operating at maximum resolution.

To accurately identify the reference point for bipolar electrode implantation, the robotic arm was vertically oriented, with its gripper and camera positioned directly downward. Once the camera verified the reference point's position, the robotic arm descended to the backpack's position. The backpack was placed within the backpack holder (Fig. 1A), yielding consistent waypoints for the robotic arm to grasp the backpack. Upon grasping the backpack, the robotic arm transported it to the same pre-implant waypoint. Next, the robotic arm conducted bipolar electrodes implantation into the exposed intersegmental membrane. Due to the constrained manipulation space available for the robotic arm ($6.5 \times 3.5 \times 2.5$ cm$^3$, between the insect and the structure), it was essential to determine an optimal implantation angle for the bipolar electrodes to avoid any potential collision. Given the symmetry of the insect's pronotum and its alignment with the reference point (Fig. 4B, iii), only the pitch angle ($\alpha$, rotation around y-axis) required adjustment while disregarding the roll and yaw angles.

During the implantation, potential collisions can occur between the backpack and the 3D-designed structure, or between the backpack branches and the insect's dorsal cuticles. Consequently, we identified pitch angles at which the backpack contacted the 3D structure ($\alpha_L$) and the insect's dorsal cuticles ($\alpha_U$), utilizing five insects. Our measurements (Fig. 4D, ii) indicated $\alpha_L$ to be $157.8° \pm 1.5°$ and $\alpha_U$ to be $167.5° \pm 2.2°$ (mean ± standard deviation). To minimize the risk of contact with either the insect or the fixation structure during implantation, an optimal mid-point angle of 162.7° was selected for the accuracy and safety of the procedure (Supplementary Movie S4).

After bipolar electrode implantation, the robotic arm applied downward force to the backpack to hook the metathorax cuticle with the backpack's four branches, completing the assembly of the hybrid robot. Subsequently, the robotic arm disengaged from the backpack and returned to its initial position to capture an image of the next fixed insect. Finally, the structure used to secure the insect was retracted, enabling the next insect to be positioned and fixed onto the platform. The entire automatic assembly, from initially fixing insect to finally releasing insect, spanned 68 s, demonstrating the effectiveness of the proposed automated assembly approach for large-scale production (Supplementary Movie S4).

**Success rate for automatic assembly.** To assess the performance of the automatic assembly system, insects were categorized based on their body length into four groups: 5.0–5.5 cm, 5.5–6.0 cm, 6.0–6.5 cm, and 6.5–7.0 cm. The success rate of assembly was then measured for each group (Table 4). The results demonstrated a notable variation in assembly success rates based on insect size. The highest success rate (86.7%) was observed in the 5.5–6.0 cm group, followed by the 5.0–5.5 cm group (80.0%). In contrast, the success rates declined markedly for larger insects, with the 6.0–6.5 cm and 6.5–7.0 cm groups achieving only 46.7% and 13.0%, respectively.

The observed variation in assembly success rates across insect size groups suggests that body dimensions significantly impact on the automatic assembly process. Instances of failure were categorized into three modes: attachment loosening, hook failure, and misalignment (Table 4, Supplementary Fig. S9). Attachment loosening occurs when the backpack mounting branches do not properly secure onto the insect's metathorax, despite partial contact (Supplementary Fig. S9A). The issue was observed only in the 5.0–5.5 cm group, accounting for 11.5% of all failures. Hook failure arises when no backpack mounting branches engage with the insect's metathorax (Supplementary Fig. S9B). For the 6.0–7.0 cm group, the success rate declined significantly due to hooking failure. Compared to the 5.5–6.0 cm group, the metathorax widths increased by 12.7% and 20.6% in the 6.0–6.5 cm and 6.5–7.0 cm groups, respectively. This increase in width hindered the secure attachment of the mounting branches onto the cuticle. Hence, hook failure was the main reason for the failed assembly trials (85.7% of the total assembly failures for 6.0–7.0 cm groups, Table 4). Misalignment occurs when the mounting device misaligns with the insect, causing only one side of the hooks to be attached to it (Supplementary Fig. S9C). Such failure mode observed in all groups except the smallest one, was mitigated in smaller insects because the system could tolerate slight positional deviations.

The high success rates of the 5.0–6.0 cm group highlight the system's precision and reliability in overcoming the challenges posed by small body dimensions and fragile insect anatomy. Combining vision-guided reference point detection for implantation, precise robotic arm manipulation, and a robust fixation structure ensured consistent bipolar electrode implantation and secure backpack attachment. These findings represent a significant advantage of automated assembly over manual assembly, eliminating the need for extensive operator skills and time when working with smaller insects[18].

The automatic assembly system surpasses manual assembly approaches. While the latter needed 15 min to assemble a single insect-computer hybrid robot[18], the former needed 68 s per insect, yielding a productivity increase of more than eleven times. The automated system achieved a success rate of 80.0–86.7% for the 5.0–6.0 cm groups. Such high success rate in these groups is because the prototype of the backpack mounting structure is based on the insects from these groups, which are majorly used for the previous studies. The implementation of an enlarged mounting structure (with a 1 mm expansion per hook branch) significantly improved the assembly success rate to 80.0% for 6.0–7.0 cm cohort (Supplementary Fig. S10), effectively mitigating the hook failure issues previously observed in larger-size groups. These results underscore the practical efficiency, reliability, and scalability of the automatic assembly system, highlighting its efficacy in the high-throughput production of insect-computer hybrid robots.

**Table 4 | Assembly success rate**

| Insect size (length, cm) | Width of metathorax (Mean ± SD, cm) | Assembly success rate | Failure modes (number of occurrences) | | |
|---|---|---|---|---|---|
| | | | Attachment loosening | Hook failure | Misalignment |
| 5.0–5.5 | 1.98 ± 0.04 | 80.0% | 3 | 0 | 0 |
| 5.5–6.0 | 2.04 ± 0.05 | 86.7% | 0 | 0 | 2 |
| 6.0–6.5 | 2.30 ± 0.14 | 46.7% | 0 | 6 | 2 |
| 6.5–7.0 | 2.46 ± 0.08 | 13.0% | 0 | 12 | 1 |

## Locomotion control and dispersion of automatically assembled insect-computer hybrid robots

To verify the controllability of the automatically assembled hybrid robots, we tested the previously established steering and stopping protocols on five such robots. The results indicated a close alignment of the performance of these automatically assembled systems with their manually assembled counterparts. The maximum steering speeds of the former were 240.0°/s and 273.5°/s for left and right turns, respectively, exhibiting a deviation of <13% compared to the latter (Fig. 5A, i). The average turning angles (Fig. 5A, ii) were 70.9° for left and 79.5° for right turns, with no significant differences detected (Student's $t$ test, $P = 0.62$ for left turns, $P = 0.50$ for right turns). The automatic assembly demonstrated consistent steering control. The difference in average turning angles between left and right was 10.8%, 50.2% lower than the manually assembled systems (21.7%). Consequently, the automatic assembly may contribute to more balanced directional control due to the prevention of manual errors caused by the assembly operator.

The deceleration decreased the average walking speed from 6.3 cm/s to 2.0 cm/s, aligning with the results observed in manually assembled systems (Student's $t$ test between automatically and manually assembled hybrid robots: $P = 0.21$). The similar deceleration performance highlights that the automatic process does not reduce control quality (Figs. 3C, iii and 5A, iii). Furthermore, the automatically assembled insect-computer hybrid robots exhibit comparable locomotion control to the manually assembled robots, validating the effectiveness and precision of the proposed process. Besides, an insect-computer hybrid robot was demonstrated to follow an S-shape line via an operator's command (Supplementary Movie S5). The path that the insect traveled aligned with the set line, which showed that the stimulation protocol and assembly strategy were successful and achieved the same level of locomotion control with the previously developed hybrid robots[47].

The automatic assembly of four insect-computer hybrid robots took 7 min and 48 s. This duration included not only the core assembly process but also additional tasks such as placing the insects on the assembly platform, removing them after assembly, placing the backpacks on the backpack holder, and initializing the robotic system operational program (Supplementary Fig. S4). These preparatory and post-assembly steps were essential to ensure smooth operation and readiness of multiple insect-computer hybrid robots. The ability to rapidly and efficiently assemble hybrid robots enhances their applicability in time-sensitive missions. To demonstrate the necessity and benefits of scalable production, this study explores terrain coverage in an unknown, obstructed outdoor environment as a fundamental task for multiple-agent systems[8].

Four automatically assembled hybrid robots were deployed onto an obstructed outdoor terrain measuring $2 \times 2$ m², with randomly placed obstacles (Fig. 5B, i). The hybrid robots were tracked using a UWB system (Fig. 5B, ii), with each hybrid robot carrying a UWB label to facilitate individual localization. Before deployment, the robots were treated with methyl salicylate on their hindleg tarsi[8] and subsequently electrically stimulated every 10 s. The trajectory of each hybrid robot was tracked (Fig. 5B, iii). The combined coverage of all four insects increased progressively, reaching 80.25% after 10 min and 31 s (Fig. 5B, iv). These findings highlight that multiple hybrid robots significantly increase the

overall coverage (80.25%; an average rate of 50.9 cm²/s) than any single insect (14.00–45.75%) (Fig. 5B, iv). Comparing the single insect's coverage (14.00–45.75%), the whole team achieved higher coverage (Fig. 5B, iv). This covering performance showed the efficiency of the simple coverage strategy using the multiple hybrid robots. While previous studies showed successful demonstrations involving multiple agents[8,13,48], our study is the first to use the insect–computer hybrid robots on an outdoor, obstructed terrain.

This study proposes an automatic assembly strategy for insect-computer hybrid robots, utilizing the discovered pronotum stimulation protocol. The effectiveness of implantation and stimulation processes was validated through locomotion studies and neural recordings from the hybrid robots. The proposed assembly method prepared hybrid robots in only 68 s, making mass production of hybrid robots a feasible endeavor. Four hybrid robots covered an outdoor terrain with a simple navigation algorithm. The result indicated the practical application of deploying multiple hybrid robots and the significance of their mass production. In the future, factories for producing insect-computer hybrid robot could be built for rapid assembly and deployment of these hybrid robots. To enhance their functionality, lightweight miniaturized thermal and RGB cameras, microphones, and IMUs can be integrated for human detection and localization, though gas sensor integration remains technically challenging due to size, power, and environmental constraints. Altogether, this work establishes a foundational platform for scalable manufacturing and real-world deployment of cyborg insects in complex, unstructured environments.

## Methods
### Insect platform
This study compared the performance of adult male Madagascar hissing cockroaches (*Gromphadorhina portentosa*, 5–7 cm, 6–8 g) with our protocol with hybrid robots[7,15] with a well-established stimulation protocol. Cockroaches were weekly provided with carrots and water inside NexGen Mouse 500 from Allen Town (25 °C, 60% relative humidity). Cockroach research was conducted with approval from the National Environmental Agency (Permit number NEA/PH/CLB/19-00012). Prior to their use in hybrid robot assembly, the cockroaches were anesthetized for 10 min under $CO_2$, and their backpacks were removed after the completion of the experiment.

### Backpack
The backpack included bipolar electrodes, a mounting structure, and a microcontroller. Below are the details for these three components. The total weight of the designed backpack is 2.3 g. As the payload for the cockroach is 15 g[49], the designed cyborg insect has another load capacity of around 12.7 g. Such remaining load capacity could accommodate more power sources or other sensing systems if needed.

**Bipolar electrodes preparation.** 3D printing offers high customization, precision, and the ability to create complex electrode structures that are difficult to achieve with mechanical machining. It also improves material efficiency and enables rapid prototyping, reducing waste and production costs[50]. Given these advantages, 3D printing was selected to optimize electrode performance and ensure experimental

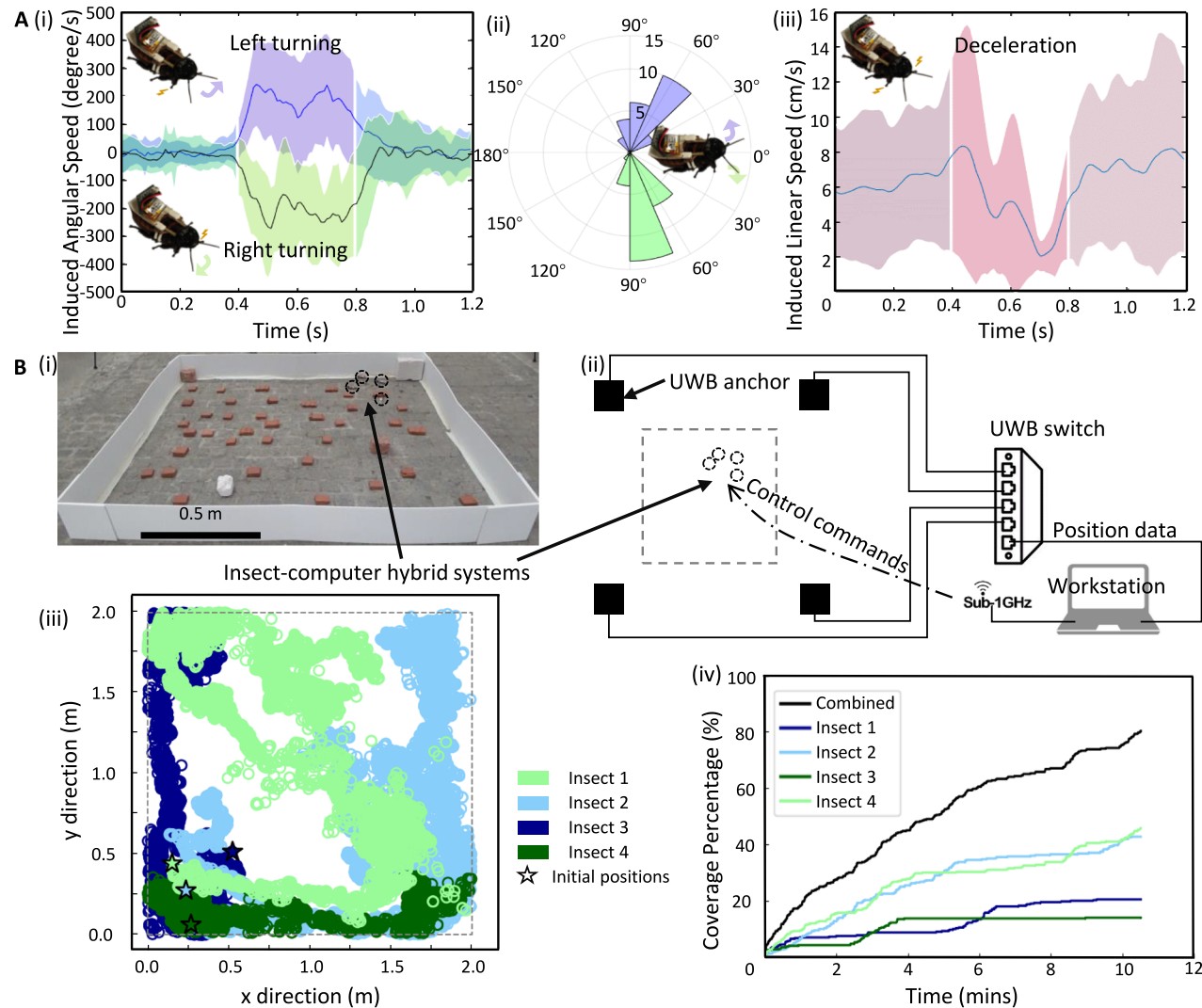

**Fig. 5 | Locomotion control and implementation of automatically assembled insect-computer hybrid robots. A** Locomotion control of insect-computer hybrid robots. i) Induced angular speed during turning stimulation (mean ± SD). ii) Angular change during turning stimulation (mean ± SD). An insignificant difference in turning angles was detected compared to manually assembled hybrid robots (two-sided Student's *t* test: *P* = 0.62 for left turns, *P* = 0.50 for right turns). iii) Induced linear speed during the deceleration stimulation (mean ± SD). An insignificant difference linear speed reduction was detected compared to manually assembled hybrid robots (two-sided Student's *t* test: *P* = 0.21). **B** Coverage of multiple insect-computer hybrid robots. i) Overview of the obstructed terrain in the coverage objective. ii) Configuration of the coverage experiment. Four UWB anchors tracked the positions of hybrid robots and transmitted the position data to the workstation for recording. Hybrid robots were controlled using a workstation through Sub-1GHz. The target region is defined within the gray dashed lines. iii) Trajectories of insect-computer hybrid robots during the mission. iv) Coverage percentage over time. The coverage rate with all four hybrid robots exceeded (80.25%) that of a single robot (14.00–45.75%), demonstrating the efficiency of multiple hybrid robots for the same coverage mission.

reproducibility. Mechanical machining should be considered for the future mass production.

**Preparation of 3D printing ink.** ABS-like photosensitive polymer raw material (SeedTech Electronics Co., Ltd.) was used in this study as an acrylonitrile butadiene styrene (ABS)−like polymer. It incorporates a light-sensitive initiator that responds to ultraviolet light (405 nm wavelength). Upon curing, cross-linking, and molding, the polymer exhibits strong mechanical properties (Supplementary Table S1).

First, 15.4 g $NH_4Cl$ was dissolved in 50 mL of deionized water. Then, 270 mg of $PdCl_2$ was added and stirred until fully dissolved to obtain 50 mL saturated activation solution with a $Pd^{2+}$ of 0.2 wt%. The solution stood for 30 min, and then 12 mL of the upper clear part was added dropwise to 38 mL of ABS-like photosensitive resin while stirring with a 1000-RPM magnetic stirrer. Finally, the ink was stirred at 1200 RPM for 30 min to yield 50 mL of active precursor, with a $Pd^{2+}$ concentration of about 0.058 wt%.

**Bipolar electrodes' fabrication with multi-material 3D printing and electroless plating.** Bipolar electrodes were fabricated using multi-material DLP 3D printing combined with selective electroless plating techniques[28–30]. The multi-material DLP 3D printing platform used a 6.1-inch 405 nm-90 W ultraviolet parallel light source of 89 MW/cm² intensity and 99% uniformity. During printing, distinct slice thicknesses and single-layer exposure times were set, including the normal resin and active precursor (Supplementary Table S2).

A composite structure of normal resin and active precursor was fabricated using a multi-material DLP 3D printing process. The active precursor's topology was selectively deposited with copper metal (Cu) during the electroless plating with nickel (Ni) substitution. The nickel-plating bath (pH = 9, 70 °C) used in this process consisted of $NiSO_4\ 6H_2O$ and $NaH_2PO_2\ H_2O$. For each printed multi-material printed part, the active precursor was distributed across the resin substrate using the designed 3D topology throughout the bipolar electrode. After the printed part was immersed in the bath, the reducing agent (sodium

**Table 5 | Policy parameters of bipolar electrode and implanted structure**

| Object | Density | Young's modulus | Poisson's ratio | Yield strength | Breaking extensibility |
|---|---|---|---|---|---|
| Bipolar electrode model | $1.26\,g/cm^3$ | 330 MPa | 0.32 | 24 MPa | 0.08 |
| Implanted structure | $1.21\,g/cm^3$ | 675 MPa | 0.38 | 38 MPa | 0.2 |

hypophosphite monohydrate) initially reduced surface-exposed palladium (Pd2+) ions to Pd monomers. These Pd monomers acted as catalytically active metal cores, initiating the ELP reaction in specific microscopic regions, and facilitating the targeted Ni metal deposition. Finally, for the copper plating (Ni substitution), a plating bath (pH = 12.2, 70 °C) consisting of $CuSO_4 \cdot 5H_2O$ and HCHO was used. The active Pd monomers were effective in initiating the electroless copper plating process.

After the electroless plating, impedance of bipolar electrodes was tested using Auto-Balancing Bridge Method and shown in the Fig. 2F. The instrument to conduct this experiment was Hioki Im3570 Impedance Analyzer.

**Finite element analysis of 3D electrodes.** Finite element computations were performed using Ansys 19.0, where a 3D bipolar electrode model was developed, material properties were defined, the mesh was generated, and the boundary conditions were set. Finite element analysis obtained the relationship between electrode implantation stress and implantation force and evaluated potential electrode damage during the process.

The finite element analysis for membrane damage simulation was conducted using the Lsdyna module in Ansys 19.0 with element type of PLANE182. A 2-mm mesh size was applied to the membrane part (Fig. 2A) to optimize the computation time while a finer 1-mm mesh was utilized for the microneedle structure of the bipolar electrodes (Fig. 2A) to improve the simulation's accuracy. The boundary conditions included a membrane (implanted structure) as fixed reference and displacement of 50 mm in the z-direction to the microneedle structure to simulate implantation. Subsequently, the simulation results were determined. The policy parameters are listed in Table 5.

**Mounting structure.** The mounting structure was securely attached to the microcontroller and bipolar electrodes. An inclined plane facilitated robotic arm's grasp, while an alignment hole ensured the consistent position of the backpack holder (Fig. 1A). Additionally, four mounting branches secured the cockroach's metathorax, with two branches on each side of the metathorax (Fig. 1C).

**Microcontroller.** The microcontroller controlling the cockroach's locomotion communicated with a workstation through Sub-1GHz frequency. Upon receiving stimulation commands from the workstation, the microcontroller outputted the electrical signals from its four stimulation channels via silver wires to the insect's target sites (Fig. 1D). Before the hybrid robot assembly, the microcontroller was stuck to the mounting structure using double-sided tapes. To avoid the silver wires being destroyed by outside obstacles and to make the backpack more compact, the stimulation channels of the microcontroller should be placed closely to the mounting structure without any silver wire drifting in the air. Hence, the microcontroller was vertically attached to the mounting structure with double-sided tapes (Steve & Leif Super Sticky). A lithium battery (3.7 V, 50 mAh) powered the microcontroller after the assembly until the commencement of the locomotion control experiment.

**Automatic assembly of insect-computer hybrid robots**
**Structure to fix insect and to expose intersegmental membrane.** To reveal the intersegmental membrane between the cockroach's pronotum and mesothorax, the pronotum was lifted from the

mesothorax. Therefore, a structure was developed with Rods A and B (Fig. 4B, iii) exerting force on the cockroach's pronotum (anterior part) and mesothorax, respectively, leveraging its posterior part. To determine the optimal lifting height $h$, of the pronotum (Fig. 4B, iv), tests were conducted on ten cockroaches, with varying fixation distances.

The middle part of the designed 3D structure was skeletonized to enable the camera to effectively capture images of insects' pronotum and detect the electrode implantation poin (Fig. 4A, iii). This skeletonized part formed a rectangle measuring $66 \times 34\,mm^2$ (Fig. 4B, i).

An anesthetized insect was positioned on the platform with its pronotum aligned with a marked spot (Fig. 4B, iii). This marked spot was placed 2 mm ahead of Rod A (Fig. 4B, iii) to ensure a secure fixation of the cockroach's protruding cuticle on the anterior pronotum. Subsequently, a slider motor drove the structure downward to execute fixation. The intersegmental membrane, located between the pronotum and mesothorax, was fully exposed, enabling the robotic arm to implant bipolar electrodes. Once successful assembly of the anesthetized insect and the attached backpack, the fixation structure was retracted (Fig. 4A, viii).

**Identification of implantation reference point based on deep learning.** Vision Transformers (ViTs) are deep learning models that have recently outperformed their CNN-based counterparts in several vision applications, including object classification and detection[51]. However, ViTs necessitate substantial datasets for effective training, such as the "Segment Anything" segmentation model, which was trained on the SA-1B dataset comprising over one billion masks and 11 million images[39]. ViTs lack locality inductive biases (correlation of image pixels and their positions); hence, their self-attention layers employ global context. In contrast, CNNs retain locality information by employing convolution layers that process images using sliding windows[52,53]. Hence, a mixture of CNN-based, ViT-based and CNN-Transformer hybrid models were surveyed for our application, to evaluate the performance of different variations of vision model architectures. The following models were trained and evaluated on our dataset of cockroaches: UNet[36], TransUNet[38], Deeplabv3[37] and Segment Anything[39], Segment Anything2[40], LM-Net[41], InceptionNeXt[42], EMCAD[43] and SHViT[44].

The UNet, Deeplabv3 and InceptionNeXt models are CNN-based models, with the Deeplabv3 model using the ResNet-101[54] and the InceptionNeXt model using ResNet-50[54], which were both pretrained on ImageNet[55], as their backbone. The TransUNet LM-Net, EMCAD and SHViT models were hybrid models, having a CNN-Transformer hybrid encoder, to combine the strengths of CNNs in locality and ViTs in the global context. The encoder for the TransUNet model combines ResNet-50[54] and ViT-B[52], which was pretrained on ImageNet[55]. The EMCAD model uses the Pyramid ViT v2 (PVTv2)[56] as its encoder, which was pretrained on ImageNet[55]. Segment Anything and Segment Anything2 were promptable foundational models, with the former using a ViT-H[52] model and the latter using a Hiera-L[57] model, and box prompts were used to specify the pronotum as the segmentation target. Apart from Segment Anything and Segment Anything2, the other models were trained on the cockroach dataset, with a batch size of 32, a two-phase learning rate scheduler, linearly raising the learning rate from 0.0001 to 0.001 during the 10 epoch warmup phase and subsequently cosine annealing was used to decay the learning after the warmup phase, Adaptive Moment Estimation as the optimizer and Binary Cross Entropy (BCE) as the loss function for 300 epochs with no early stopping condition.

Overall, 29 unique cockroaches were fixed with the custom-designed 3D structure. The robotic arm was placed in a fixed position to capture images with the Intel RealSense D435 camera. We used $256 \times 256$-pixel crop from the original image, centered around the pronotum as the input for training the models. This approach shortened the training and inference times. Of the collected images, 20 were used as test cases for model evaluation, while the remaining were used for training. Training data were augmented to enhance model robustness against variances in pronotum rotation and shape. Asymmetrical scaling of the training samples in the x- and y- axes, using scaling values between 0.8 and 1.2, was applied to the training samples using bilinear interpolation to generate more unique pronotum shapes to simulate the varying pronotum sizes of cockroaches. Subsequently, rotations between $-6\,°$ and $6\,°$ were applied to accommodate inconsistencies in positions when cockroaches were mounted on the 3D structure. The final training dataset contained 6570 images after data augmentation, and furthermore an 80/20 train/validation split was used, leading to 5254 images used for training and 1316 images used for validation.

A data ablation study was conducted, where the performance of the models was compared when trained on four conditions: only original unaugmented images (9 images), original images with augmentation using asymmetrical scaling in x-y axes (1316 images), original images with augmentation using rotation (90 images) and the full augmented training dataset (6570 images).

**Automatic manipulation of the robotic arm.** After the cockroach was fixed and its intersegmental membrane exposed, the Intel RealSense D435 camera, mounted on the Robotiq Hand e gripper captured images of the cockroach through the skeletonized 3D structure (Fig. 4A, iii, B, i). Using a pretrained model, a reference point was identified on the middle posterior edge of the pronotum (green point in Fig. 4A, iii). Since the camera provided the depth information, the x, y, z positions of the reference point relative to the robotic arm's base were identified following hand-eye calibration with the UR3e robotic arm. Subsequently, the robotic arm grasped the backpack, which was always placed on the backpack holder (Fig. 1A).

The backpack's bipolar electrodes were precisely implanted within the exposed intersegmental membrane (Fig. 4A, v), by programming the robotic arm to prevent collisions with the cockroach or the fixation structure. Since the implantation reference position ($p_R$) was predetermined, the implantation angle became a critical parameter. The cockroach's alignment with the predetermined mark and its symmetrical pronotum simplified the implantation process, focusing only on the pitch angle. The robotic arm grasped the backpack and positioned the bipolar electrodes' tips under the pronotum at varying pitch angles until the backpack touched the 3D structure (lower threshold, $\alpha_L$) or the cockroach (upper threshold, $\alpha_U$). Data for these thresholds were collected from $N = 5$ cockroaches.

The bipolar electrodes were implanted into the insect's intersegmental membrane using the implantation reference point and pitch angle. Next, the backpack was pressed down until its mounting branches hooked the cockroach's metathorax (Fig. 4A, vi). Finally, the gripper released the backpack (Fig. 4A, vii) and the robotic arm returned to its initial state, enabling the camera to capture an image of the next cockroach.

**Success rate for automatic assembly.** Twenty cockroaches were divided into four groups based on their body sizes. Each cockroach was automatically assembled thrice to ensure system robustness and consistency. An attempt was considered successful if the bipolar electrodes were implanted into the intersegmental membrane and the backpack was securely fixed to the metathorax, with no detachment after completion. The success rate for each body size group was computed by dividing the number of successful assembly attempts by the total number of attempts ($3 \times 5 = 15$). All experiments had identical robotic arm settings, camera calibration, and fixation configuration, to secure comparability of the results.

## Neural recording during the stimulation

We recorded and assessed insects' neural responses to the electrical stimulation to determine the optimal stimulation strength. Three cockroaches were anesthetized with $CO_2$ for 10 min, after which their ventral nerve cords were exposed through neck dissection. The bipolar electrode was implanted in the intersegmental membrane between the pronotum and mesothorax to transmit electrical stimulation generated by the backpack's microcontroller (a single bipolar square-wave pulse of 1 Hz and 0.5–4.0 V amplitude for 1.0 s). Each stimulation was repeated thrice. The nerve cords were rinsed with cockroach saline for visibility under a microscope. Two probes were fixed to the nerve cord to record the transmitted signals and a ground pin was implanted into the cockroach's abdomen.

Neural responses recorded during the electrical stimulation included some influence of electrical signals. Therefore, neural signals, starting at 0, 0.5, and 1.0 s (the pulse edges), were set to zero for the first 50 ms. Subsequently, the neural signals were filtered using a second-order Butterworth filter (300–5000 Hz), and neural spikes were detected with a threshold $T$.

$$T = 5 \times median(|x|/0.6745)$$

where $x$ signifies the filtered signals. The detected neural spikes are indicated by blue circles (Fig. 3A, ii), which were quantified at varying stimulation voltages (Fig. 3A, iii).

## Locomotion control of insect-computer hybrid robots

Five insect-computer hybrid robots, both manually and automatically assembled, were tested for locomotion control. Electrical stimulation using a bipolar pulse wave (0.4 s, 3.0 V, 42 Hz) was applied to stimulate the insects. Each stimulation type (right/left turn and deceleration) was repeated five times. Insects' locomotion responses were recorded using a motion tracking system (VICON).

## Dispersion of multiple insect-computer hybrid robots

Four insect-computer hybrid robots were rested for four hours after the assembly. They were then deployed for the coverage task on obstructed terrain ($2.0 \times 2.0$ m², Fig. 5B, i). Four UWB anchors were positioned at the corners of a $3.6 \times 3.6$ m² area (Fig. 5B, ii) to track the hybrid robots equipped with UWB labels. Hybrid robots were released from the designated area's corner. Before their release, a chemical booster, methyl salicylate (Sigma-Aldrich)[8] was applied to the hindleg tarsi of the insects. Such chemical was proven effective to improve insects' motion activeness level for covering mission. Application of this chemical aimed to increase the movement activeness level of the hybrid robots, thereby facilitating better terrain coverage. After release, the robots were stimulated randomly (steered or decelerated) to explore the terrain. Random electrical stimulation type was chosen to simulate a decentralized, autonomous exploration strategy, which reflects real-world scenarios where multiple agents operate without a pre-defined navigation path. This approach allows for unbiased coverage distribution and reduces dependency on precise localization or predefined control algorithms. The terrain was divided into 400 squares, each measuring $10 \times 10$ cm² ($S_{square}$) for easy coverage computation. Any hybrid robot passing through a particular square deemed that region covered. The number of covered squares is noted as $n_{covered}$. The covered area ($S$) and coverage rate ($C$) were calculated as below,

$$S = n_{covered} \times S_{square} \tag{1}$$

$$C = \Delta S / \Delta t \tag{2}$$

where $n_{covered}$ is the number of covered squares, $\Delta S$ is the change of the covered area, and $\Delta t$ is the change of time (Supplementary Fig. S11). The trajectory sampling rate is 20 Hz.

## Reporting summary

Further information on research design is available in the Nature Portfolio Reporting Summary linked to this article.

## Data availability

All data supporting the findings of this study are available within the article and its supplementary files. Source data are provided with this paper.

## Code availability

The code that supports the findings of this study is available within the Supplementary Materials files.

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

## Acknowledgements

The authors thank Mr. See To Yu Xiang, Dr. Kazuki Kai, Dr Duc Long Le, Mr. Li Rui for their suggestions, and Ms. Kerh Geok Hong, Wendy, for her support and help. This study was funded by JST (Moonshot R&D Program, Grant Number JPMJMS223A, H.S.).

## Author contributions

Conceptualization: Q.L., H.S. Investigation: Q.L., N.V. Methodology: Q.L., K.S., P.T.T.N., G.A.G.N. Visualization: Q.L., K.S. Funding acquisition: H.S. Supervision: H.S., Writing – original draft: Q.L., K.S., G.A.G.N., Writing – review & editing: Q.L., N.V., K.S., P.T.T.N., G.A.G.N., H.S.

## Competing interests

The authors declare no competing interests.
