## [Transparent Peer Review file · Nature Communications]

Cyborg Insect Factory: Automatic Assembly for Insect-computer Hybrid Robot via Vision-guided Robotic Arm Manipulation of Custom Bipolar Electrodes

Corresponding Author: Dr Qifeng Lin

Version 0:

Reviewer comments:

Reviewer #1

(Remarks to the Author)

The authors successfully demonstrate the automatic assembly of insect-computer hybrid robots/ cyborg insects using a robotic arm guided by a deep-learning vision system. This is a significant step towards the mass production of these robots for various applications.

The authors make several significant contributions to this work:

1. Automated Assembly of Cyborg Insects:

Previous research relied heavily on manual electrode implantation, a time-consuming and labor-intensive process. The authors introduce a novel approach utilizing a robotic arm guided by deep learning vision, enabling automated assembly of cyborg insects. This automation enhances efficiency in the production process, paving the way for the potential mass production of cyborg insects.

2. Custom Bipolar Electrodes:

The development of specialized bipolar electrodes with microneedle structures and secure hooks represents a key innovation. The use of 3D printing and electroless plating allows for precise fabrication and optimization of electrode properties.

3. UWB-based Localization:

Traditional motion tracking systems face limitations in outdoor environments and when dealing with obstacles. The authors demonstrate the successful application of Ultra-Wideband (UWB) technology for localizing cyborg insects. UWB provides robust tracking capabilities, even in challenging conditions, making it suitable for real-world applications like search and rescue.

4. Deceleration Control:

This study presents the first successful implementation of deceleration control in insect-computer hybrid robots. By electrically stimulating specific areas of the insect's body, the authors achieve a significant reduction in walking speed. This crucial control mechanism enhances the maneuverability and operational capabilities of cyborg insects.

These innovations collectively advance the field of insect-computer hybrid robots/cyborg insects, bringing us closer to the realization of practical applications in various domains/applications.

However, some points need clarification from the authors.

- abstract:

"The proposed automatic assembly strategy reduced preparation time for the insect-computer hybrid robots while maintaining their precise control ..."

In terms of precise control, could the authors demonstrate the precise control of the cyborg insect? for example, the authors could show this claim by controlling the cyborg insect autonomously or manually.

- Page 3, line 68

"However, insects' tiny and delicate body structures made the manual surgery process time-consuming and difficult." Should the authors mention the time required to make a cyborg insect and what is its difficulty in the Introduction?

- Page 8, line 188

"The reduction in neural activity at 4.0 V may be due to damage to the insects' neural system", could the authors provide citations/references for this statement?

- Page 10, line 231

"During the real application, deceleration helped the insect-computer hybrid robot to avoid the collision after detecting its front obstacles [ref. 32]".

Based on reference 32, I could not find the deceleration control in this paper, instead left/right turn and forward motion were used for the obstacle avoidance as depicted in Fig. 6. Could the authors clarify this statement?

- For the deceleration control, I found that ref [1, and 2] used left/right turn and forward motion in locomotion control for the obstacle avoidance system. It would be better if the authors could provide some details/discussion on the benefit of left/right turn and deceleration motion compared to left/right turn and forward motion in terms of locomotion control for example obstacle negotiation.

[ref 1] Li, R., Lin, Q., Tran-Ngoc, P. T., Le, D. L., & Sato, H. (2024). Smart insect-computer hybrid robots empowered with enhanced obstacle avoidance capabilities using onboard monocular camera. *npj Robotics*, 2(1), 2.

[ref 2] Ariyanto, M., Refat, C. M. M., Yamamoto, K., & Morishima, K. (2024). Feedback control of automatic navigation for cyborg cockroach without external motion capture system. *Heliyon*, 10(5).

- For motion control the authors proposed, the authors proposed three kinds of locomotion, i.e. turn left, turn right, and deceleration control. How about the acceleration control? Is deceleration control more important in the locomotion control of cyborg insects?

could three kinds of locomotion proposed by authors be used to control the cyborg insect in complex environments for example goal-based navigation/obstacle avoidance?

- While the authors mention why the antennae and abdomen are unsuitable for electrode implantation in their automated system, they could more explicitly state the advantages of using the pronotum. Highlighting the pronotum's size, accessibility, and suitability for automated manipulation would strengthen their argument.

- Fig. 1, the authors attach the wireless backpack vertically. What is the consideration to attach it vertically instead of horizontally? Does it affect the locomotion of cyborg insects in complex environments or challenging terrain?

- What kind of sensor that has been integrated to the proposed backpack? what is the capability of this backpack in the locomotion control? any sensing capability?

- In autonomous navigation (Fig. 5) as demonstrated in Fig.5 utilized obstacles without corners. Obstacle having a corner could be challenging for navigating cyborg insects (cockroaches) as shown by the author's previous study [ref 32] and they like to walk toward the corner. What the authors thought about this?

- What is the solution for the Hook failure on the backpack?

- Page 16 line 391

"In contrast, the automatic process achieved 68 seconds for assembly of one insect, delivering more than eleven times higher productivity of the previous manual preparation while maintaining success rates of 80.0–86.7% for the 5.0 – 6.0 cm size range." what is the success rate for manual assembly?

- Page 17, line 411

"The reason of this reduction was that the automatic assembly avoided manual errors from the assembly operator". Could the authors provide the details of the error in manual assembly?

- Section 4.6: The authors could clarify the role of methyl salicylate in the covering mission, and why it is selected as chemical stimulation.

Reviewer #2

(Remarks to the Author)

The authors designed an automatic assembly method for insect-computer hybrid robots. The method accomplished the backpack mounting task with precise implantation of bipolar electrodes using a robotic arm. A deep learning-based vision system was used to locate the implantation sites. The automatically assembled hybrid robots revealed similar performances with manually assembled systems. The proposed automatic assembly strategy reduced preparation time for the insect-computer hybrid robots. Overall, the paper demonstrates a meaningful work on insect-computer hybrid robots, which will reduce the mass-producing time and cost for practical applications. Thus, I think this paper is appropriate for publication after making some revisions.

Please follow the comments below.

1. The exoskeleton of the pronotum is thick and rigid. How do the authors decide the size, structure and stiffness of the needle to reliably puncture the exoskeleton?

2. What is the benefit of 3D printing bipolar electrodes? Why not choose some other ways to make the electrodes. For example, the mechanical machining seems a more cost-effective. I suggest the authors discuss more on the advantages of the electrode.
3. From line 237 to 240, the authors discussed the standard deviation of speed. However, I believe it is not proper to compare the standard deviation here because the average speed is different. The standard deviation might be different under different speeds even without stimulation. Moreover, I think the distribution of speeds does not follow normal distribution when the speed is slow, which is because the speed can not be negative.
4. In section 2.3.1, the variables, e.g., d and h , should be italic.
5. In Fig. 3A(ii), the time scale of the ENG should be marked in the figure.
6. It seems the mounting hooks have the uniform size. Can this approach deal with cockroaches of different sizes?
7. Locations of the obstacles should be marked on Fig.5B(iii).
8. In line 391, the authors mentioned the assembly time is 68 s. In line 422, the assembly time of 4 robots is 7 min 48 s in total. I think the differences in assembly time should be provided.
9. The authors used a 6DoF robotic arm for implantation. Is it possible to use a simpler system to implement, such as SCARA? What is the advantage of the 6DoF robotic arm? Will it influence the precision?

Reviewer #3

(Remarks to the Author)

****Summary****

This paper introduces the pioneering concept of an "Insect Robot Factory," a truly captivating notion. It automates the electrode implantation and backpack fixation processes for cyborg insects, allowing them to decelerate and steer under controlled conditions. Consequently, the feasibility of multi-agent collaboration is validated. Specifically, this study introduces an automated assembly system that employs a robotic arm and computer vision-based site localization for insect-computer hybrid robots. This system facilitates the installation of a backpack and the precise implantation of custom-designed bipolar electrodes. Additionally, a stimulation protocol was developed targeting the intersegmental membrane between the pronotum and mesothorax of the Madagascar hissing cockroach, enabling steering and deceleration control performance. Based on this, a multi-agent system comprising four hybrid robots successfully navigates outdoor terrain. This work lays the foundation for animal robot mass production and practical application, enhancing production efficiency and advancing the field.

In light of this, the reviewer recommends that the article be accepted for publication before addressing the comments below:

****Advantage****

1. This paper presents an automatic cockroach robot assembly system, which provides the possibility of mass production. The efficiency assembly process takes only 68 seconds and indicates the scalable fabrication for insect-computer hybrid robots.
2. This paper introduces a novel cockroach stimulation paradigm that successfully achieves orientation and deceleration control of an insect-computer hybrid robot.
3. The performance of the Cockroach robot is verified in detail in three steps. They are the experiment of assembly efficiency, control effect and cooperative navigation coverage.
 - a. This paper presents a comprehensive analysis of the automated assembly process across various insect sizes. The highest success rate was observed in the 5.5–6.0 cm group (86.7%), followed by the 5.0–5.5 cm group (80.0%). These findings provide valuable insights into the method's reliability and effectiveness for small insects.
 - b. This paper presents experiments on the control effects of steering and deceleration control. They illustrate the effectiveness of the cockroach assembled automatically. Electrical stimulation prompted the insects to turn left at an average angle of 68.0 degrees and to turn right at an average angle of 82.6 degrees. The maximum angular speeds reached 275.8 degree/s for left turns and 298.2 degree/s for right turns. The deceleration control process takes 0.33 seconds to reduce the average walking speed of the insects from 6.2 cm/s to a minimum of 1.5 cm/s.
 - c. Multiple cockroach robots are deployed to collaborate in completing the area coverage task. The coverage rate achieved is 80.25% in 10 minutes and 31 seconds, demonstrating that the efficiency of multiple cockroach robots is significantly improved compared to that of a single cockroach robot (less than 50% shown as in Fig. 5B(iii)).

****Questions****

1. Concerns regarding the section for automatic implantation of electrodes and backpack:
 - a) Could you please provide the exact coordinates and error range of the stimulation site of the cockroach robot with the reference point as the origin?
 - b) When the backpack shown in Fig.1B,C is mounted on a cockroach, how to adjust the width between the electrodes according to the body size of the cockroach? In other words, how to ensure that the system can accurately implant the electrodes into the stimulation site when facing different sizes of cockroaches?
 - c) What is the advantage for the chip on the backpack being mounted vertically? Will this orientation prevent cockroaches from navigating some of the small tunnels they could otherwise traverse?
 - d) How much is the load capacity of the proposed cyborg robot?
 - e) In Figure.3A(iii), you have presented the experimental curve of the stimulation voltage and the corresponding number of spikes. Please supplement the specific process of this experiment. Additionally, at the beginning of Section 2.2, you stated the selection of a 3.0V voltage. Please explain the principle of the relationship between the stimulation voltage magnitude and the number of spikes, or provide references.
 - f) It is mentioned in Figure 1.D that the battery has already been connected to the backpack, but in Figures 1B and 1C, the position where the battery is mounted by the robotic arm is not shown. Please provide additional information on how the battery is stably fixed to the cockroach's back.

g) Please provide the entire assembly process flowchart, especially the order of battery installation. Please provide the detailed parameters, weight, and volume of the backpack, as well as the stable operating time under the current battery power supply.

2. Concerns regarding the experiments :

a) What should be done if the cockroach robot is reluctant to move during navigation? The stimulation paradigm presented in this paper includes steering and deceleration commands but does not incorporate forward commands.

b) What are the motivations and advantages of choosing to send control commands randomly in the experiment?

c) What is the unique function of the proposed deceleration instruction in the arena shown in Fig. 5B(i) that contributes to the coverage improvement? Can specific terrain be designed in the arena to highlight the effect of the deceleration command?

d) Please give the formula for calculating the coverage rate, as well as the key parameters such as the coverage radius and the trajectory sampling rate when the cockroach is in a certain position. And add the parameter legend in Fig. 5B(iii).

e) It is recommended to include control and experimental groups for both non-stimulated and stimulated motion traces, with parameters including distance traveled, head speed, and time spent.

f) Please supplement the results of coverage experiments using one, two, and three cockroaches, each conducted at least five times, and report the mean and standard deviation of coverage. Thus, the relationship between the number of cockroach robots and the coverage rate is more convincing.

g) The proposed method for direction control through stimulation of the intersegmental membrane in the thorax of cockroaches has not been directly compared with existing antenna-based and abdominal stimulation methods in terms of performance metrics, including the success rate of stimulation, response times, and the physiological impact of long-term stimulation on the cockroaches.

h) How long is the lifespan of the cockroaches implanted using this method compared to those implanted manually by hand. If possible, could you provide additional data on this comparison?

i) Section 2.4 reports that assembling four insect-computer hybrid robots took 7 minutes and 48 seconds, whereas Section 2.3.4 indicates that the automatic process requires 68 seconds per insect. This results in a total assembly time for four robots that is approximately 6.88 times longer than for a single robot, suggesting potential scalability issues. It is recommended that the authors provide further clarification and analysis on the time efficiency of assembling multiple hybrid robots consecutively to address this discrepancy.

3. Concerns regarding the preparation of electrodes :

a) A subsequent electroless plating process is conducted after multi-material 3D printing to improve its corrosion resistance (Fig. 2B). More data, e.g. comparison on surface morphology by means of optical images or scanning electron microscope, are expected to illustrate the resistance of electrochemical corrosion of the electrode under stimulation cycles with the variation of voltage.

b) Fig. 2F shows the impedance characteristics of bipolar electrodes, which was stably below 70 Ω . It is recommended to add detailed steps and instruments used in the impedance test. Besides, the corresponding phase changes with frequency are supposed to be provided to illustrate the comprehensive electrochemical properties of electrodes.

c) In Table 4, the material parameters are defined for the finite element analysis of implantation stress. It would be better to provide the element type for the FEA model.

4. Some opinions on writing and pictures:

a) The primary focus of this paper is the cockroach control paradigm and the implantation of an automatic robotic arm. Therefore, could these two components be presented in Section II instead of the content related to electrodes?

b) There are some images that remain too blurry, even in the largest and highest definition versions. For example, the legend on the left side of each figure in Fig. 2D is unclear. Additionally, Fig. 3B is also blurry.

c) The reviewer commented on the language of the manuscript, noting that it contains grammatical and spelling errors that impact its clarity and readability. These linguistic issues must be addressed.

d) The organization and logic of the manuscript require careful refinement. The abstract centers on the automatic assembly system for insect robots developed in this study, while the introduction highlights the discovery of directional control sites based on the stimulation of the intersegmental membrane between the pronotum and the mesothorax, with only a brief mention of the assembly system at the end. This structure may lead readers to believe that the authors have not clearly articulated the contributions and logical progression of the paper.

5. About the deep learning-based Detection of the Pronotum part:

a) In the section on the Detection of Pronotum using deep learning, the paper evaluates segmentation models such as UNet, Deeplabv3, TransUNet, and Segment Anything, all of which are methods from or before 2023. Given the emergence of new models and architectures in the field after SAM, these should also be assessed.

b) The amount of training and testing data in the Detection of Pronotum is unclear; the authors mention experiments with 29 unique cockroaches but do not specify the number of images in the training and testing sets.

c) Additionally, the paper does not detail the quantity of data after augmentation or quantify the impact of using and not using data augmentation on the performance of the segmentation models.

Version 1:

Reviewer comments:

Reviewer #1

(Remarks to the Author)

The authors addressed the reviewers' comments by conducting additional experiments, providing clarifications, and revising the manuscript. This included clarifying locomotion control in the abstract, detailing the manual assembly process and challenges, providing explanations and citations for observed neural activity, correcting misleading statements about

deceleration control, explaining the vertical backpack attachment, addressing hook failure issues, and clarifying the role of methyl salicylate in the covering mission. These changes improved the manuscript's clarity, completeness, and persuasiveness.

There are some minor suggestions for improvements:

- If possible, the authors could provide a more detailed discussion on how deceleration control complements steering in obstacle avoidance. A comparative discussion or comments with existing methods (e.g., forward motion + steering or deceleration motion + steering) would strengthen the author's argument.
- Section 2.4 describes the locomotion control experiments. While the authors have added a section on the S-shape line following, it lacks details about the control algorithm or strategy used. Including this information would enhance the reproducibility of the study.
- The manuscript mentions potential sensor integration but lacks specifics. The authors could outline feasible sensor types (e.g., gas sensors, thermal array for search-and-rescue, etc) and their impact on payload capacity.

Visuals:

- Some figures, like Fig. 2D, are noted to be blurry. High-resolution versions of all images should be used for clarity.
- Consider adding scale bars to all relevant microscopy images to provide a clear size context.
- Fig. 5B(iii): Please add a color legend for the trajectory plot and indicate the initial position(s) of the cyborg insects. Same as Fig. S12.

Reviewer #2

(Remarks to the Author)

In the second review, I think the authors have addressed most of my concerns. The manuscript has become more convincing with newly added details and experiments. The proposed automatic assembly strategy is an innovative work on insect-computer hybrid robots, which will reduce the mass-producing time and cost. The novelty and completeness meet the requirement of the journal. Thus, I think this manuscript is suitable for publication on Nature Communications after making minor revisions.

Please follow the comments below.

1. The authors mentioned that the electrode implantation sites located in the intersegmental membrane beneath the anterior thoracic dorsal plate. Is there any variability in the mechanical properties of the intersegmental membrane across individuals or different life cycle stages? Are such individual variations likely to affect the consistency of electrode implantation or the effectiveness of stimulation?
2. The authors mentioned that the success rate of assembly varied among insects with different body sizes. The 6.5-7.0 cm group only achieved the success rate of 13%, which is much lower than the 5.5-6.0 cm group. Can the authors discuss more on the possible reasons? Is it possible to solve it by improving the automatic assembly strategy?

Reviewer #3

(Remarks to the Author)

The author has addressed most of my concerns and added the requested experiments and corresponding content. However, there are still some minor problems as follows. I recommend acceptance after minor revisions.

R3-3: Does the width of the bipolar electrode only have to be less than 1.2mm? Does it need to be greater than a minimum?

R3-8: Please provide the specific model numbers and electrical parameters (including operating voltage and power consumption) of the backpack electronic components, along with battery specifications and corresponding operational duration. It is recommended to include a schematic diagram of the final battery configuration in the carried device.

R3-9: How is methyl salicylate distributed in the arena? Is it uniformly distributed, only at the starting point, or does it have a stepwise distribution? At what point do you consider choosing such a distribution?

R3-11: Can you provide some specific examples, such as some motion sequence clips from the navigation task in Fig. 5B(iii), to illustrate the role of the deceleration command in sharp turns, dead-end paths, or highly confined regions?

R3-12: Please provide the specific calculation method of covered space, that is, the radius of the covered area of each cockroach at each time point.

REVIEWER COMMENTS

Reviewer #1:

[R1-1] The authors successfully demonstrate the automatic assembly of insect-computer hybrid robots/ cyborg insects using a robotic arm guided by a deep-learning vision system. This is a significant step towards the mass production of these robots for various applications.

The authors make several significant contributions to this work:

1. Automated Assembly of Cyborg Insects:

Previous research relied heavily on manual electrode implantation, a time-consuming and labor-intensive process. The authors introduce a novel approach utilizing a robotic arm guided by deep learning vision, enabling automated assembly of cyborg insects. This automation enhances efficiency in the production process, paving the way for the potential mass production of cyborg insects.

2. Custom Bipolar Electrodes:

The development of specialized bipolar electrodes with microneedle structures and secure hooks represents a key innovation. The use of 3D printing and electroless plating allows for precise fabrication and optimization of electrode properties.

3. UWB-based Localization:

Traditional motion tracking systems face limitations in outdoor environments and when dealing with obstacles. The authors demonstrate the successful application of Ultra-Wideband (UWB) technology for localizing cyborg insects. UWB provides robust tracking capabilities, even in challenging conditions, making it suitable for real-world applications like search and rescue.

4. Deceleration Control:

This study presents the first successful implementation of deceleration control in insect-computer hybrid robots. By electrically stimulating specific areas of the insect's body, the authors achieve a significant reduction in walking speed. This crucial control mechanism enhances the maneuverability and operational capabilities of cyborg insects.

These innovations collectively advance the field of insect-computer hybrid robots/cyborg insects, bringing us closer to the realization of practical applications in various domains/applications.

However, some points need clarification from the authors.

[Reply to this comment]

Thank you for reviewing the manuscript. We have fully read all your comments and conducted more experiments and investigations. We responded to all your questions in the following pages. Thanks to your advice, the revised manuscript has a clearer explanation and is much more convincing.

[R1-2] - abstract:

"The proposed automatic assembly strategy reduced preparation time for the insect-computer hybrid robots while maintaining their precise control ..."

In terms of precise control, could the authors demonstrate the precise control of the cyborg insect? for example, the authors could show this claim by controlling the cyborg insect autonomously or manually.

[Reply to this comment]

We understand that "precise" control is misleading. Hence, we clarified the description of locomotion control in the abstract and emphasized that the newly developed cyborg insect is as the same level with previously developed ones^{1,2} in terms of control effect. According to your comments, we demonstrated a cyborg insect to follow an S-shape line (length: 1.0 m) to show the controllability of our designed cyborg insect (Movie S5). The observed control performance aligns with prior studies conducted by Tahmid et al. on the cockroach² and Nguyen et al. on the beetle³. The mean deviation between the trajectory of the cyborg insect and the predefined line is 0.9 cm, further validating its controllability.

[Changes in manuscript]

Revised the abstract to highlight quantitative performance metrics.

Line Number 43-50.

The automatically assembled hybrid robots demonstrated steering control (over 70 ° for 0.4 s stimulation) and deceleration control (68.2% speed reduction for 0.4 s stimulation), achieving comparable locomotion control performance to manually assembled systems (Student's t-test: $P = 0.62$ for left turns, $P = 0.50$ for right turns, $P = 0.21$ for deceleration). To demonstrate the control effect, a hybrid robot was controlled to follow an S-shape line. The path taken by the hybrid robot highly overlaps with the predefined line.

The following part was added to the Section 2.4.

Line Number 434-437.

Besides, an insect-computer hybrid robot was demonstrated to follow an S-shape line (Movie S5). The path that the insect travelled aligned with the set line, which showed that the stimulation protocol and assembly strategy were successful and achieved the same level of locomotion control with the previously developed hybrid robots².

[R1-3] - Page 3, line 68

"However, insects' tiny and delicate body structures made the manual surgery process time-consuming and difficult." Should the authors mention the time required to make a cyborg insect and what is its difficulty in the Introduction?

[Reply to this comment]

We added details on the time required for manual assembly and its associated challenges to the Introduction. This contextualizes the efficiency improvement achieved by the proposed automatic assembly. The difficulty for making a cyborg insect includes two aspects. First aspect is that the insect's target implantation sites have small size. For example, insect's cerci are ~ 0.6 mm in diameter and antennae is less than 1 mm. Meanwhile, the silver wire commonly used is 70 μm in diameter⁴. All the electrodes and target implantation sites are smaller than 1 mm, which requires high accuracy for the detection and implantation if automatic assembly is considered. Besides, second aspect is that the implantation sites, i.e., antennae and cerci, are soft organs. Their shape changes during the surgery, which causes difficulty even for human operator (Fig. R1-1). Such soft target implantation sites also make it impossible for their fixation by mechanical structure even if automatic assembly is considered.

[Changes in manuscript]

Fig. R1-1 was added to Page 3 of Supplementary Materials, as Fig. S1.

The following part was added to the Introduction part.

Line Number 69-74.

However, the intricate and fragile anatomy of insects makes the manual surgery process time-consuming (~ 15 minutes per insect⁵) and difficult⁶. For instance, insects' small and soft antennae and cerci necessitate the use of specialized microinstruments and microscopes to manipulate their tissues accurately. Even minimal force can cause unintended shape changing, making procedures highly demanding (Fig. S1).

Fig. R1-1. Conventional implantation on the cockroach's antennae and cerci. (A) Silver wire implantation on the antennae. (i) Before implantation. (ii) Silver wire started inserting the cut antenna.

(iii) Antennae changed shape during the silver wire inserting. (B) Silver wire implantation on the cerci. (i) Before implantation. (ii) Silver wire started inserting the cut cerci. (iii) Cerci shrunk during the silver wire inserting.

[R1-4] - Page 8, line 188

"The reduction in neural activity at 4.0 V may be due to damage to the insects' neural system", could the authors provide citations/references for this statement?

[Reply to this comment]

The possible neural damage of high stimulation amplitude was previously discussed in a study on neural circuit recording from an intact cockroach nervous system⁷. Titlow et al. advised against using voltages much higher than the maximal threshold for recruitment, "There is no reason to go to voltages much higher than maximal threshold for recruitment as a high voltage can be damaging to the nerve."

Besides the cockroach, there are also studies indicating that neural responses to electrical stimulation can initially increase with rising stimulation amplitude but may decrease if the amplitude continues to increase beyond a certain point. This phenomenon suggests that while moderate levels of electrical stimulation can enhance neural activity, excessively high amplitudes may lead to a reduction in neural responses.

For instance, Alejandro et al. examined the effects of high-amplitude electrical stimulation on neuronal activity⁸. They found that while increasing the stimulation amplitude initially enhanced neuronal responses, further increases led to a reduction in elicited neuronal activity. They concluded that "high-amplitude electrical stimulation can reduce elicited neuronal activity," highlighting the importance of optimizing stimulation parameters to avoid diminished neural responses.

Also, research on intracortical microstimulation (ICMS) has demonstrated that while the magnitude of neuronal responses is linearly correlated with stimulation amplitude up to a certain point, further increases can lead to a steady state or even a reduction in response⁹. This suggests that beyond a certain threshold, additional increases in stimulation amplitude do not elicit stronger neural responses and may instead induce mechanisms that suppress activity.

These findings underscore the complex relationship between stimulation parameters and neural responses, emphasizing the need for careful calibration to achieve desired outcomes without inducing adverse effects.

[Changes in manuscript]

The following part was added to the Section 2.2.

Line Number 188-190.

However, increasing the stimulation voltage to 4.0 V yielded a 23.5% decline in the average number of spikes, denoting reduced neural activity due to potential damage to the insects' neural system⁷⁻⁹.

[R1-5] - Page 10, line 231

"During the real application, deceleration helped the insect-computer hybrid robot to avoid the collision after detecting its front obstacles [ref. 32]".

Based on reference 32, I could not find the deceleration control in this paper, instead left/right turn and forward motion were used for the obstacle avoidance as depicted in Fig. 6. Could the authors clarify this statement?

[Reply to this comment]

Sorry for misleading. In the reference 32, only forward and steering stimulation were used. In this reference, when the dead-end paths happened, insects naturally moved forward and then was trapped inside the corner (illustrated in Fig. 2 of that paper). In this situation, deceleration control developed from our studies could help avoid the insects moving forward and being trapped. For the locomotion control of the insect-computer hybrid robot, there are two aspects for their control, i.e., direction control and speed control. For the direction control, left and right turning stimulation can realize insect's direction changing. For the speed control, both acceleration and deceleration should be considered. However, previous studies only talked about the acceleration electrical stimulation and could not realize the deceleration control⁶. Previously, we discovered that chemical stimulation stably increased insects' activeness level, which replaced the function of forward electrical stimulation. Hence, in this study, to control the speed more fully, we considered to introduce the deceleration and combined the chemical stimulation for the real application. Thank you.

[Changes in manuscript]

The following part was added to the Section 2.2.

Line Number 226-231.

For the control of the insect-computer hybrid robot, both direction control and speed control are fundamental. Previously designed direction control, including left and right turning speed realized the full direction control of the insect-computer hybrid robot. However, for the speed control, only acceleration stimulation was realized. Hence, our discovery on the deceleration control could realize the speed control more fully.

[R1-6] - For the deceleration control, I found that ref [1, and 2] used left/right turn and forward motion in locomotion control for the obstacle avoidance system. It would be better if the authors could provide some details/discussion on the benefit of left/right turn and deceleration motion compared to left/right turn and forward motion in terms of locomotion control for example obstacle negotiation. [ref 1] Li, R., Lin, Q., Tran-Ngoc, P. T., Le, D. L., & Sato, H. (2024). Smart insect-computer hybrid robots empowered with enhanced obstacle avoidance capabilities using onboard monocular camera. *npj Robotics*, 2(1), 2.

[ref 2] Ariyanto, M., Refat, C. M. M., Yamamoto, K., & Morishima, K. (2024). Feedback control of automatic navigation for cyborg cockroach without external motion capture system. *Heliyon*, 10(5).

[Reply to this comment]

Thank you very much for your comment. Based on these two references, even without acceleration stimulation, the cockroach still moved forward. This is because insects' nature of forward moving. Their developed forward stimulation is mainly used to increase insect's moving speed temporarily. However, such speed increase was also achieved by other stimulation protocol. For example, in our previous study¹⁰, we discovered a chemical booster which stably increased insects' moving speed. Hence, electrical forward stimulation is replaceable. Different from the acceleration stimulation, insect's deceleration has not been developed so far. Therefore, in this study, we developed the deceleration control for the insect-computer hybrid robot and combined with chemical booster for the real application. Meanwhile, in the coverage mission, as the repeated steering stimulation led to less responses¹¹, combining the deceleration stimulation prevented such inactive responses.

[Changes in manuscript]

The following part was added to the Section 2.2.

Line Number 223-231.

Furthermore, the cockroach decelerated when the electrical stimulation was outputted from the outer electrodes of the bipolar electrode pair (Fig. 2A), making the first successful implementation of the deceleration control in insect-computer hybrid robots. For the control of the insect-computer hybrid robot, both direction control and speed control are fundamental. Previously designed direction control, including left and right turning speed realized the full direction control of the insect-computer hybrid robot. However, for the speed control, only acceleration stimulation was realized. Hence, our discovery on the deceleration control could realize the speed control more fully.

[R1-7] - For motion control the authors proposed, the authors proposed three kinds of locomotion, i.e. turn left, turn right, and deceleration control. How about the acceleration control? Is deceleration control more important in the locomotion control of cyborg insects? could three kinds of locomotion proposed by authors be used to control the cyborg insect in complex environments for example goal goal-based navigation/obstacle avoidance?

[Reply to this comment]

We understand acceleration control has been developed widely to increase insect's moving speed. Both acceleration and deceleration are important for the real application. Acceleration helps insects with higher motion activeness level. Deceleration control was necessary for the insects with high moving speed to avoid collision with obstacles especially in the obstructed terrain. However, high motion activeness can also be achieved by other stimulation sources rather than electrical stimulation, e.g., chemical stimulation¹². Hence, our electrical stimulation protocol used for deceleration control rather than acceleration. For the navigation, the direction control strategies, i.e., left and right stimulation are the main functions used. A cyborg insect was used for demonstration on a S-shape line following. Such navigation is also presented in other research^{2,3}, which showed that our stimulation protocol can also achieve same level of navigation even without acceleration stimulation.

[Changes in manuscript]

Movie S5 was added to show the S-shape line following.

The following part was added to the Introduction.

Line Number 434-437.

Besides, an insect-computer hybrid robot was demonstrated to follow an S-shape line (Movie S5). The path that the insect travelled aligned with the set line, which showed that the stimulation protocol and assembly strategy were successful and achieved the same level of locomotion control with the previously developed hybrid robots².

[R1-8] - While the authors mention why the antennae and abdomen are unsuitable for electrode implantation in their automated system, they could more explicitly state the advantages of using the pronotum. Highlighting the pronotum's size, accessibility, and suitability for automated manipulation would strengthen their argument.

[Reply to this comment]

We clarified our rationale for selecting the pronotum as the implantation site. In addition to discussing the limitations of the antennae and abdomen, we highlight the advantages of using the pronotum. Specifically, the pronotum offers a relatively large surface, facilitating precise detection, fixation and electrode implantation. We incorporate these points into the revised manuscript to strengthen our argument.

[Changes in manuscript]

The following part was added to the Introduction.

Line Number 108-113.

Notably, the pronotum cuticle is both larger and thicker than the abdominal cuticles, measuring approximately 11.6 - 13.4 mm in length, 0.5 - 0.6 mm in thickness, which eases attachment and detection. Additionally, akin to the interspace between abdominal segments¹³, an intersegmental membrane between the pronotum and the mesothorax is the key target site for stimulation, enabling directional control of the insect (Fig. 1C).

[R1-9] - Fig. 1, the authors attach the wireless backpack vertically. What is the consideration to attach it vertically instead of horizontally? Does it affect the locomotion of cyborg insects in complex environments or challenging terrain?

[Reply to this comment]

Thank you for your enlightening questions. The vertical attachment of the wireless microcontroller is securer for wiring of the silver wires, making the backpack more compact and later automatic assembly easier. In the complex environment, the wiring is especially important as it may be damaged by the sharp obstacles. Besides, regarding the effect on the cyborg insect's locomotion, the microcontroller weighs only 1.4 g (Table R1-1), while the rest of the hybrid system, including the insect and the

backpack without the microcontroller, weighs 7.0 – 9.0 g. Given this weight distribution, the centre of mass of the entire system is minimally affected by the microcontroller's orientation. Hence, the microcontroller vertical attachment does not change the locomotion control of the cyborg insects in complex terrain.

[Changes in manuscript]

Table R1-1 was added to Page 17 of Supplementary Materials, as Table S3.

The following part was added to the Section 4.2.3.

Line Number 564-568.

To avoid the silver wires being destroyed by outside obstacles and to make the backpack more compact, the stimulation channels of the microcontroller should be placed closely to the mounting structure without any silver wire drifting in the air. Hence, the microcontroller was vertically attached to the mounting structure with double-sided tapes (Steve & Leif Super Sticky).

Table R1-1. Detailed parameters of the backpack

Backpack elements	Length (mm)	Width (mm)	Height (mm)	Volume (mm ³)	Weight (g)
Mounting structure	32.0	21.4	14.6	575.5	0.8
Bipolar electrode	20.7	5.0	1.6	31.3	0.05
Microcontroller	15.0	15.0	17.0	787.5	1.4

[R1-10] - What kind of sensor that has been integrated to the proposed backpack? what is the capability of this backpack in the locomotion control? any sensing capability?

[Reply to this comment]

The present backpack is used for outputting electrical stimulation and communication between the central station and cyborg insects. It does not include additional sensors. As the load capacity of the cockroach is 15 g¹⁴ and current backpack is only 2.3 g, the cyborg insect can be adapted with sensors for environmental sensing and autonomous navigation if needed. We have clarified that point in the revised manuscript. Thank you for the question.

[Changes in manuscript]

The following part was added to the Section 4.2.

Line Number 490-493.

The total weight of the designed backpack is 2.3 g. As the payload for the cockroach is 15 g¹⁴, the designed cyborg insect has another load capacity of around 12.7 g. Such remaining load capacity could accommodate more power sources or other sensing systems if needed.

[R1-11] - In autonomous navigation (Fig. 5) as demonstrated in Fig.5 utilized obstacles without corners. Obstacle having a corner could be challenging for navigating cyborg insects (cockroaches) as shown by the author's previous study [ref 32] and they like to walk toward the corner. What the authors thought about this?

[Reply to this comment]

Thank you very much for your valuable questions. To investigate the unknown environment, our strategy includes two steps: (1) disperse the hybrid robots and to cover the environment, (2) navigate the hybrid robots to the uncovered terrain. This study focused on (1) while we focused on (2) in [ref 32], where we use camera to navigate the insect away from the corner and to approach the target.

[R1-12] - What is the solution for the Hook failure on the backpack?

[Reply to this comment]

Thank you very much for your valuable questions. Hook failure happened during the assembly (Fig. S10). As discussed in the Section 2.3.4, the hook failure mainly happened for the larger size insects. That is because we developed our prototype based on the small size insects, which are majorly used for the present researches^{5,15-19}. The failure resulted from the insect's larger size could be avoided by some potential design improvements, like size-specific hooks. For instance, we enlarged the design of the hook structure by 1 mm for each branch, and the assembly success rate increased to 80.0% for 6.0 – 7.0 cm group (Fig. S11).

[Changes in manuscript]

Fig. R1-2 was added to Page 13 of Supplementary Materials, as Fig. S11.

The following part was added to the Section 2.3.4.

Line Number 403-408.

The automated system achieved a success rate of 80.0%–86.7% for the 5.0 – 6.0 cm groups. Such high success rate in these groups is because the prototype of the backpack mounting structure is based on the insects from these groups, which are majorly used for the previous studies. The enlarged mounting structure (1 mm enlarged for each hook branch) increased the assembly success rate for 6.0-7.0 cm groups to 80.0% (Fig. S11).

Fig. R1-2. Assembly for large size of cockroach (6.0–7.0 cm) with different sizes of mounting structures. The enlarged mounting structure increased success rate of automatic assembly from 46.7% (6.0 – 6.5 cm) and 13.0% (6.5 – 7.0 cm) to 80.0% (6.0 – 7.0 cm). (A) Hook failure with normal size of mounting structure. (B) Success with enlarged size of mounting structure.

[R1-13] - Page 16 line 391

"In contrast, the automatic process achieved 68 seconds for assembly of one insect, delivering more than eleven times higher productivity of the previous manual preparation while maintaining success rates of 80.0–86.7% for the 5.0 – 6.0 cm size range." what is the success rate for manual assembly?

[Reply to this comment]

Thank you for your valuable question. Manual assembly, when performed by a skilled operator with careful adjustments, generally results in a high success rate. This is because the operator can make real-time judgments and modifications during the process. However, manual assembly is time-consuming, requiring approximately 15 minutes per insect. In contrast, our newly designed bipolar electrodes simplify the implantation procedure, making it more suitable for robotic operation while maintaining a consistently high success rate.

[R1-14] - Page 17, line 411

"The reason of this reduction was that the automatic assembly avoided manual errors from the assembly operator". Could the authors provide the details of the error in manual assembly?

[Reply to this comment]

As the antennae and cerci, which are commonly used for target stimulation sites of manual assembly for cyborg insects, are soft, it's difficult for the operator to implant electrodes directly without any organ shape changing (Fig. R1-1). Hence, the manual errors during the implantation happen frequently. However, even the error happens a lot, the operator can always modify and adjust the electrode implantation direction to comprise the error. Therefore, manual assembly is very time-consuming. And that is the main point for us to develop the automatic assembly strategy.

[Changes in manuscript]

Fig. R1-1 was added to Page 3 of Supplementary Materials, as Fig. S1.

The following part was added to the Introduction part.

Line Number 71-74.

For instance, insects' small and soft antennae and cerci necessitate the use of specialized microinstruments and microscopes to manipulate their tissues accurately. Even minimal force can cause unintended shape changing, making procedures highly demanding (Fig. S1).

[R1-15] - Section 4.6: The authors could clarify the role of methyl salicylate in the covering mission, and why it is selected as chemical stimulation.

[Reply to this comment]

Based on our previous discovery, methyl salicylate effectively increased the coverage for covering mission¹⁰. Hence, in this study, we directly utilized it to increase the coverage. We elaborated more on the role of methyl salicylate as a locomotory booster and its effectiveness in enhancing insect movement. We agree that illustrating more on this part helps the readers understand more easily. Thank you for the suggestion.

[Changes in manuscript]

The following part was added to the Section 4.6.

Line Number 708-712.

Hybrid robots were deployed in the designated area's corner with a chemical booster, methyl salicylate¹². Application of this chemical aimed to increase the movement activeness level of the hybrid robots, thereby facilitating better terrain coverage. After release, the robots were stimulated randomly (steered or decelerated) to explore the terrain.

Reviewer #2:

[R2-1] The authors designed an automatic assembly method for insect-computer hybrid robots. The method accomplished the backpack mounting task with precise implantation of bipolar electrodes using a robotic arm. A deep learning-based vision system was used to locate the implantation sites. The automatically assembled hybrid robots revealed similar performances with manually assembled systems. The proposed automatic assembly strategy reduced preparation time for the insect-computer hybrid robots. Overall, the paper demonstrates a meaningful work on insect-computer hybrid robots, which will reduce the mass-producing time and cost for practical applications. Thus, I think this paper is appropriate for publication after making some revisions.

Please follow the comments below.

[Reply to this comment]

Thank you very much for your enlightening advice. We have learned a lot from your comments. Based on your suggestions, quality of our manuscript has been improved significantly. We will respond to each of your comments below.

[R2-2] The exoskeleton of the pronotum is thick and rigid. How do the authors decide the size, structure and stiffness of the needle to reliably puncture the exoskeleton?

[Reply to this comment]

Thank you for pointing out the structure issue in the view of pronotum. We also agree that the exoskeleton of the pronotum is difficult to penetrate. Thus, we do the electrode implantation to the intersegmental membrane below the pronotum. In this case, the exoskeleton of the pronotum is intact after the electrode implantation. To reliably puncture the intersegmental membrane, we verified the feasibility of the designed electrodes using finite element simulations and performed mechanical tests to ensure the needles penetrated the membrane without damage. These findings were included in Section 2.2. We understand it is easier for the readers to show the intersegmental membrane more clearly to avoid the misunderstanding. Hence, we added a picture for the intersectional membrane as Fig. S9.

[Changes in manuscript]

Fig. R2-1 was added to Page 11 of Supplementary Materials, as Fig. S9.

Fig. R2-1. Stimulation sites with error of implantation. As the origin detection has the error of 1.749 pixels, the real error for the implantation is 0.36 mm.

[R2-3] What is the benefit of 3D printing bipolar electrodes? Why not choose some other ways to make the electrodes. For example, the mechanical machining seems a more cost-effective. I suggest the authors discuss more on the advantages of the electrode.

[Reply to this comment]

3D printing offers significant advantages, including high customization, precision in fabrication, and the ability to create complex geometries that are difficult to achieve with mechanical machining. Below are the advantages of 3D printing which are important to this study.

First, customization and design flexibility: 3D printing allows for highly customized electrode designs that can be easily adjusted to accommodate insect morphologies and experimental requirements. Unlike mechanical machining, which may be limited by tool constraints, additive manufacturing enables the creation of intricate structures with precise control over dimensions and material distribution.

Second, fabrication precision and repeatability: The layer-by-layer deposition process in 3D printing ensures high precision in electrode fabrication, reducing variability between samples. This is particularly important for ensuring consistency in electrical stimulation and minimizing potential deviations caused by manufacturing tolerances.

Third, complex geometries and integrated structures: 3D printing facilitates the production of intricate electrode shapes that may be challenging or impossible to achieve with mechanical machining. For instance, in this study, it allows for the direct integration of bipolar electrodes, customized holes for mounting structure, and specific hook design to fix the electrodes inside the insect body, thereby improving contact stability and biocompatibility.

Finally, material efficiency and rapid prototyping: Additive manufacturing reduces material waste compared to subtractive machining processes. Moreover, it enables rapid prototyping and iterative design improvements without requiring expensive molds or specialized machining setups, making it a time- and cost-effective solution for research and development.

We agree that mechanical machining is more cost-effective. In this study, we focused on the rapid prototyping. Once the design settled down, we can transfer to mechanical machining in the future. Thank you for the suggestion.

To better address this point, we have added a discussion in Section 4.2.1 of the revised manuscript highlighting these benefits and explaining why 3D printing was selected for electrode fabrication in this study.

[Changes in manuscript]

The following part was added to the Section 4.2.1.

Line Number 495-500.

3D printing offers high customization, precision, and the ability to create complex electrode structures that are difficult to achieve with mechanical machining. It also improves material efficiency and enables rapid prototyping, reducing waste and production costs²⁰. Given these advantages, 3D printing was selected to optimize electrode performance and ensure experimental reproducibility. Mechanical machining should be considered for the future mass production.

[R2-4] From line 237 to 240, the authors discussed the standard deviation of speed. However, I believe it is not proper to compare the standard deviation here because the average speed is different. The standard deviation might be different under different speeds even without stimulation. Moreover, I think the distribution of speeds does not follow normal distribution when the speed is slow, which is because the speed cannot be negative.

[Reply to this comment]

Instead of discussing the standard deviation of speed, we normalized it with its mean value. Thank you for pointing out that issue. With your kind help, the quality of our manuscript is improved.

[Changes in manuscript]

The following part was added to the Section 2.2.

Line Number 235-238.

The standard deviation of the minimum speed normalized with the mean speed during stimulation was 0.83. This small normalized standard deviation in the minimum decelerated speed highlights the consistency of the deceleration stimulation.

[R2-5] In section 2.3.1, the variables, e.g., *d* and *h*, should be italic.

[Reply to this comment]

We have corrected the formatting of the variables throughout the whole manuscript accordingly.

[R2-6] In Fig. 3A(ii), the time scale of the ENG should be marked in the figure.

[Reply to this comment]

We have included time scale accordingly. Thank you.

[Changes in manuscript]

Fig. 3A(ii) included time scale.

[R2-7] It seems the mounting hooks have the uniform size. Can this approach deal with cockroaches of different sizes?

[Reply to this comment]

The current mounting hooks were designed based on the cockroaches within a specific size range, i.e., 5.0 – 6.0 cm in length. This is because the insects in this size are the majority and also used for other studies^{5,15-19}. For broader applicability to insects from 6.0-7.0 cm groups, 1-mm enlarged hooks effectively avoided the hook failure (Fig. R2-2) and increased the success rate to 80.0%.

[Changes in manuscript]

Fig. R2-2 was added to Page 13 of Supplementary Materials, as Fig. S11.

The following part was added to the Section 2.3.4.

Line Number 403-408.

The automated system achieved a success rate of 80.0%–86.7% for the 5.0 – 6.0 cm groups. Such high success rate in these groups is because the prototype of the backpack mounting structure is based on the insects from these groups, which are majorly used for the previous studies. The enlarged mounting structure (1 mm enlarged for each hook branch) increased the assembly success rate for 6.0-7.0 cm groups to 80.0% (Fig. S11).

Fig. R2-2. Assembly for large size of cockroach (6.0–7.0 cm) with different sizes of mounting structures. The enlarged mounting structure increased success rate of automatic assembly from 46.7% (6.0 – 6.5 cm) and 13.0% (6.5 – 7.0 cm) to 80.0% (6.0 – 7.0 cm). (A) Hook failure with normal size of mounting structure. (B) Success with enlarged size of mounting structure.

[R2-8] Locations of the obstacles should be marked on Fig.5B(iii).

[Reply to this comment]

We understand including detailed information of the obstacles inside the Fig.5B (iii) makes the mapping situation easier to understand. However, to show the real situation of the obstacles, we directly used the picture of the tested terrain in Fig.5B (i), which is also easy for the readers to have an intuitive impression on the terrain situation. For Fig.5B (iii), we used it only for showing the trajectories of the insect-computer hybrid robots.

[R2-9] In line 391, the authors mentioned the assembly time is 68 s. In line 422, the assembly time of 4 robots is 7 min 48 s in total. I think the differences in assembly time should be provided.

[Reply to this comment]

Thank you for pointing out this discrepancy. The reported 68 seconds per insect refers strictly to the robotic automatic assembly process, which includes fixing and releasing insect, implanting electrodes and securing the backpack. In contrast, the 7 minutes and 48 seconds required for assembling four robots accounts for additional tasks, such as placing the insects on the assembly platform, removing them after assembly, and initializing the robotic system operational program. These preparatory and post-assembly steps are necessary to ensure smooth operation to prepare the multiple cyborg insects. However, such additional tasks are also necessary for manual assembly⁵, e.g., placing and removing insect under the microscope, placing electrodes/tweezers besides the microscope. And such tasks are

also not included in the 15 minutes from the manual assembly in the previous study⁵. To fairly compare the assembly duration, 68 s for the automatic assembly procedure was only considered in this study. We have clarified this distinction in the revised manuscript to better reflect the realistic workflow and time allocation for consecutive assembly tasks.

[Changes in manuscript]

Fig. R2-3 was added to Page 7 of Supplementary Materials, as Fig. S5.

The following part was added to the Section 2.3.4.

Line Number 438-447.

The automatic assembly of four insect-computer hybrid robots took 7 min and 48 s. This duration included not only the core assembly process but also additional tasks such as placing the insects on the assembly platform, removing them after assembly, placing the backpacks on the backpack holder, and initializing the robotic system operational program (Fig. S5). These preparatory and post-assembly steps were essential to ensure smooth operation and readiness of multiple insect-computer hybrid robots. The ability to rapidly and efficiently assemble hybrid robots enhances their applicability in time-sensitive missions. To demonstrate the necessity and benefits of scalable production, this study explores terrain coverage in an unknown, obstructed outdoor environment as a fundamental task for multiple-agent systems¹².

Fig. R2-3. Flowchart of the preparation for multiple insect-computer hybrid robots.

[R2-10] The authors used a 6DoF robotic arm for implantation. Is it possible to use a simpler system to implement, such as SCARA? What is the advantage of the 6DoF robotic arm? Will it influence the precision?

[Reply to this comment]

Thank you for suggesting the simpler system. We understand SCARA may have advantages over the cost. However, in this study, we need to optimize the implantation pitch angle of the bipolar electrodes. In this case, we need the robotic arm to be flexible to adjust the orientation of the end effector. SCARA only has one rotation freedom, which is hard to quickly adjust the orientation of the end effector (Fig. R2-4A). Unlike SCARA robots, the 6DoF robotic arm allows for more freedom in terms of the ability to change the position and orientation of the end effector (Fig. R2-4B). This capability is particularly crucial for adapting to the three-dimensional structure of the insect body, ensuring that the electrodes can be inserted at the optimal angles with minimal disturbance to the insects' unrelated body parts. Furthermore, based on our testing with the UR3e robotic arm, the system demonstrated high precision in assembling insect-computer hybrid robots, maintaining consistent implantation accuracy. Nevertheless, considering the cost-effectiveness of the SCARA, it can be considered once the prototyping has been finished and ready for mass production.

[Changes in manuscript]

Fig. R2-4 was added to Page 4 of Supplementary Materials, as Fig. S2.

Fig. R2-4. Comparison between SCARA and 6DOF robot. End effector of SCARA robot only has one freedom of rotation and cannot change the pitch angle flexibly. However, end effector of 6DOF robot has three freedoms of rotation. During the study of the implantation pitch angle, 6DOF robot is more suitable.

Reviewer #3:

[R3-1] **Summary******

This paper introduces the pioneering concept of an "Insect Robot Factory," a truly captivating notion. It automates the electrode implantation and backpack fixation processes for cyborg insects, allowing them to decelerate and steer under controlled conditions. Consequently, the feasibility of multi-agent collaboration is validated. Specifically, this study introduces an automated assembly system that employs a robotic arm and computer vision-based site localization for insect-computer hybrid robots. This system facilitates the installation of a backpack and the precise implantation of custom-designed bipolar electrodes. Additionally, a stimulation protocol was developed targeting the intersegmental membrane between the pronotum and mesothorax of the Madagascar hissing cockroach, enabling steering and deceleration control performance. Based on this, a multi-agent system comprising four hybrid robots successfully navigates outdoor terrain. This work lays the foundation for animal robot mass production and practical application, enhancing production efficiency and advancing the field. In light of this, the reviewer recommends that the article be accepted for publication before addressing the comments below:

******Advantage******

1. This paper presents an automatic cockroach robot assembly system, which provides the possibility of mass production. The efficiency assembly process takes only 68 seconds and indicates the scalable fabrication for insect-computer hybrid robots.

2. This paper introduces a novel cockroach stimulation paradigm that successfully achieves orientation and deceleration control of an insect-computer hybrid robot.

3. The performance of the Cockroach robot is verified in detail in three steps. They are the experiment of assembly efficiency, control effect and cooperative navigation coverage.
 - a. This paper presents a comprehensive analysis of the automated assembly process across various insect sizes. The highest success rate was observed in the 5.5–6.0 cm group (86.7%), followed by the 5.0–5.5 cm group (80.0%). These findings provide valuable insights into the method's reliability and effectiveness for small insects.

 - b. This paper presents experiments on the control effects of steering and deceleration control. They illustrate the effectiveness of the cockroach assembled automatically. Electrical stimulation prompted the insects to turn left at an average angle of 68.0 degrees and to turn right at an average angle of 82.6 degrees. The maximum angular speeds reached 275.8 degree/s for left turns and 298.2 degree/s for right turns. The deceleration control process takes 0.33 seconds to reduce the average walking speed of the insects from 6.2 cm/s to a minimum of 1.5 cm/s.

 - c. Multiple cockroach robots are deployed to collaborate in completing the area coverage task. The coverage rate achieved is 80.25% in 10 minutes and 31 seconds, demonstrating that the efficiency of multiple cockroach robots is significantly improved compared to that of a single cockroach robot (less than 50% shown as in Fig. 5B(iii)).

[Reply to this comment]

Thank you for your time on reviewing our manuscript and providing insightful comments. In this response, we have addressed your questions point-by-point and have revised the manuscript accordingly. We hope you find the revised manuscript to be clearer, and more convincing.

****Questions****

1. Concerns regarding the section for automatic implantation of electrodes and backpack:

[R3-2] Could you please provide the exact coordinates and error range of the stimulation site of the cockroach robot with the reference point as the origin?

[Reply to this comment]

We provided the coordinates and error range of stimulation sites with the reference point as the origin accordingly (Fig. R3-1).

[Changes in manuscript]

Fig. R3-1 was added to Page 11 of Supplementary Materials, as Fig. S9.

Fig. R3-1. Stimulation sites with error of implantation. As the origin detection has the error of 1.749 pixels, the real error for the implantation is 0.36 mm.

[R3-3] When the backpack shown in Fig.1B,C is mounted on a cockroach, how to adjust the width between the electrodes according to the body size of the cockroach? In other words, how to ensure that the system can accurately implant the electrodes into the stimulation site when facing different sizes of cockroaches?

[Reply to this comment]

Thank you for your comment. In this study, we developed our prototype of the backpack based on the majority of the cockroach, which has the size of 5.0 – 6.0 cm and commonly studied^{5,15–19}. The width of the membrane across cockroaches with different sizes was measured (1.4 ± 0.2 cm). To ensure that the bipolar electrodes inserted inside the intersegmental membrane, the width of the bipolar electrodes was set as 1.0 cm to satisfy the minimum width, i.e., 1.2 cm. Meanwhile, the accuracy of the robotic arm is 0.03 mm^{21} , which is much lower than the possible implantation failure, 0.36 mm error (Fig. R3-1). Hence, our developed system accurately implanted the electrodes.

[Changes in manuscript]

Fig. R3-1 was added to Page 11 of Supplementary Materials, as Fig. S9.

The following part was added to the Section 2.2.

Line Number 179-182.

The width of the intersegmental membrane between pronotum and mesothorax was measured (1.4 ± 0.2 cm) across cockroaches with different sizes. To ensure that the bipolar electrodes implanted into the membrane, the distance between the two bipolar electrodes was set as 1.0 cm to satisfy minimum width of the membrane, i.e., 1.2 cm.

[R3-4] What is the advantage for the chip on the backpack being mounted vertically? Will this orientation prevent cockroaches from navigating some of the small tunnels they could otherwise traverse?

[Reply to this comment]

The vertical attachment of the wireless microcontroller is easier for wiring of the silver wires, making the backpack more compact and later automatic assembly easier. As the presence of the sub-1 GHz antenna (Height is 17 mm), the height of the cyborg insect did not change much with the different attachment direction of the wireless microcontroller ($15 \times 15 \text{ mm}^2$). Hence, the different orientations of the microcontroller attachment should not make difference for the cockroach movement in the small tunnels. Thank you for the question.

[Changes in manuscript]

The following part was added to the Section 4.2.3.

Line Number 559-570.

The microcontroller controlling the cockroach's locomotion communicated with a workstation through Sub-1 GHz frequency. Upon receiving stimulation commands from the workstation, the microcontroller outputted the electrical signals from its four stimulation channels via silver wires to the insect's target sites (Fig. 1D). Before the hybrid robot assembly, the microcontroller was stuck to the mounting structure using double-sided tapes. To avoid the silver wires being destroyed by outside obstacles and to make the backpack more compact, the stimulation channels of the microcontroller should be placed closely to the mounting structure without any silver wire drifting in the air. Hence, the microcontroller was vertically attached to the mounting structure with double-sided tapes (Steve & Leif Super Sticky).

A lithium battery (3.7 V, 50 mAh) powered the microcontroller after the assembly until the commencement of the locomotion control experiment.

[R3-5] How much is the load capacity of the proposed cyborg robot?

[Reply to this comment]

We added more information on the cyborg insect's load capacity accordingly. The payload capacity for the selected insect platform is approximately 15 g¹⁴. In our current design, the total weight of the backpack, including all necessary electronic components, is 2.3 g. Therefore, the remaining **usable load capacity** of the cyborg insect is around 12.7 g, which could potentially accommodate additional functional modules, such as extended power sources or supplementary sensing systems. Thank you for pointing out the concern on the load capacity. Adding this information helps readers understand more deeply about our cyborg insect.

[Changes in manuscript]

The following part was added to the Section 4.2.

Line Number 490-493.

The total weight of the designed backpack is 2.3 g. As the payload for the cockroach is 15 g¹⁴, the designed cyborg insect has another load capacity of around 12.7 g. Such remaining load capacity could accommodate more power sources or other sensing systems if needed.

[R3-6] In Figure.3A(iii), you have presented the experimental curve of the stimulation voltage and the corresponding number of spikes. Please supplement the specific process of this experiment. Additionally, at the beginning of Section 2.2, you stated the selection of a 3.0V voltage. Please explain the principle of the relationship between the stimulation voltage magnitude and the number of spikes, or provide references.

[Reply to this comment]

We have provided the specific process for the neural recording experiment in Section 4.4.

The selection of a 3.0 V stimulation voltage was based on balancing effective neural activation with minimizing potential neural damage. The relationship between stimulation voltage magnitude and the number of spikes follows a well-documented nonlinear trend: increasing voltage generally enhances neural firing up to a certain threshold, beyond which further increases may lead to diminished responses or potential neural damage.

Previous studies on cockroach nervous systems have highlighted the risks associated with excessive stimulation amplitudes. One study on neural circuit recordings from intact cockroach nervous systems cautioned against using voltages significantly higher than the maximal recruitment threshold, stating

that high voltages can be damaging to nerves⁷. This principle has also been observed in other neurostimulation studies, where an initial increase in stimulation amplitude leads to enhanced neuronal activity, but excessive voltages may reduce spike counts^{8,9}.

These findings indicate that optimal stimulation voltage selection is crucial to achieving effective neural activation while avoiding overstimulation-induced suppression or damage. Based on these principles, we selected 3.0V as an appropriate voltage to reliably elicit neural spikes while minimizing the risk of excessive stimulation effects.

[Changes in manuscript]

The following part was added to the Section 2.2.

Line Number 188-192.

However, increasing the stimulation voltage to 4.0 V yielded a 23.5% decline in the average number of spikes, denoting reduced neural activity due to potential damage to the insects' neural system⁷⁻⁹. To prevent unnecessary damage and maintain effective stimulation, a 3.0 V voltage was considered optimal for the subsequent discussions.

The following part was added to the Section 2.2.

Line Number 679-697.

We recorded and assessed insects' neural responses to the electrical stimulation to determine the optimal stimulation strength. Three cockroaches were anesthetized with CO₂ for 10 min, after which their ventral nerve cords were exposed through neck dissection. The bipolar electrode was implanted in the intersegmental membrane between the pronotum and mesothorax to transmit electrical stimulation generated by the backpack's microcontroller (a single bipolar square-wave pulse of 1 Hz and 0.5 – 4.0 V amplitude for 1.0 s). Each stimulation was repeated thrice. The nerve cords were rinsed with cockroach saline for visibility under a microscope. Two probes were fixed to the nerve cord to record the transmitted signals and a ground pin was implanted into the cockroach's abdomen.

Neural responses recorded during the electrical stimulation included some influence of electrical signals. Therefore, neural signals, starting at 0, 0.5, and 1.0 s (the pulse edges), were set to zero for the first 50 ms. Subsequently, the neural signals were filtered using a second-order Butterworth filter (300–5000 Hz), and neural spikes were detected with a threshold T .

$$T = 5 \times \text{median} (|x|/0.6745)$$

where x signifies the filtered signals. The detected neural spikes are indicated by blue circles (Fig. 3A, ii), which were quantified at varying stimulation voltages (Fig. 3A, iii)

[R3-7] It is mentioned in Figure 1.D that the battery has already been connected to the backpack, but in Figures 1B and 1C, the position where the battery is mounted by the robotic arm is not shown. Please provide additional information on how the battery is stably fixed to the cockroach's back.

[Reply to this comment]

Thank you for pointing this out. In the early research on the preparation of cyborg insect, the assembly between the insect, electrodes and microcontroller is the most challenging task and time-consuming (15 mins)⁵. The battery plugging inside the microcontroller socket is not a complex task and does not take time. Also, insects after the assembly needed time for recovery as they were anathemized and took surgeries. Hence, battery was only attached when the cyborg insect was used for experiment. As a result, we study the automatic assembly without the battery. After the assembly finished and the hybrid robot was needed for the locomotion control, the battery was plugged inside the power socket. Double-sided tapes were stuck to the battery first so that the battery was fixed to the backpack.

[Changes in manuscript]

The following part was added to the Fig.1 Caption.

Line Number 908-910.

After the assembly finished and the hybrid robot was needed for the locomotion control, a LiPo battery was plugged inside the power socket. Double-sided tapes were used to stick the battery to the backpack.

[R3-8] Please provide the entire assembly process flowchart, especially the order of battery installation. Please provide the detailed parameters, weight, and volume of the backpack, as well as the stable operating time under the current battery power supply.

[Reply to this comment]

Accordingly, we provide the flowchart (Fig. R3-2) and information about the backpack (Table R3-1). For the power supply, based on our testing, the stable operating time is about 20 mins for mapping experiment. Thank you for the suggestion.

[Changes in manuscript]

Table R3-1 was added to Page 17 of Supplementary Materials, as Table S3.

Fig. R3-2 was added to Page 7 of Supplementary Materials, as Fig. S5.

The following part was added to the Section 2.3.4.

Line Number 438-447.

The automatic assembly of four insect-computer hybrid robots took 7 min and 48 s. This duration included not only the core assembly process but also additional tasks such as placing the insects on the assembly platform, removing them after assembly, placing the backpacks on the backpack holder, and initializing the robotic system operational program (Fig. S5). These preparatory and post-assembly steps were essential to ensure smooth operation and readiness of multiple insect-computer hybrid robots. The ability to rapidly and efficiently assemble hybrid robots enhances their applicability in time-sensitive missions. To demonstrate the necessity and benefits of scalable production, this study explores terrain coverage in an unknown, obstructed outdoor environment as a fundamental task for multiple-agent systems¹².

Table R3-1. Detailed parameters of the backpack

Backpack elements	Length (mm)	Width (mm)	Height (mm)	Volume (mm ³)	Weight (g)
Mounting structure	32.0	21.4	14.6	575.5	0.8
Bipolar electrode	20.7	5.0	1.6	31.3	0.05
Microcontroller	15.0	15.0	17.0	787.5	1.4

Fig. R3-2. Flowchart of the preparation for multiple insect-computer hybrid robots.

2. Concerns regarding the experiments:

[R3-9] What should be done if the cockroach robot is reluctant to move during navigation? The stimulation paradigm presented in this paper includes steering and deceleration commands but does not incorporate forward commands.

[Reply to this comment]

Thank you for your pointing out the possibility of the insect's low activeness level. How to keep the insects with high activeness level is very important for their real application. Actually, we also considered this issue in this paper. In our demonstration for mapping, chemical stimulation with methyl salicylate was used to keep insects with high activeness level to keep insect moving. Such stimulation protocol to keep insect with high moving activeness level was discovered and verified in our previous study¹⁰.

[Changes in manuscript]

The following part was added to the Section 4.6.

Line Number 708-712.

Hybrid robots were deployed in the designated area's corner with a chemical booster, methyl salicylate¹². Application of this chemical aimed to increase the movement activeness level of the hybrid robots, thereby facilitating better terrain coverage. After release, the robots were stimulated randomly (steered or decelerated) to explore the terrain.

[R3-10] What are the motivations and advantages of choosing to send control commands randomly in the experiment?

[Reply to this comment]

To explore an unknown terrain, our strategy includes two steps. The first step is to increase the coverage by multiple insect-computer hybrid robots. With our developed automatic assembly system, we can prepare the hybrid robots shortly. As some parts of the target terrain may still be undiscovered by the hybrid robots, our next step is to navigate the hybrid robots to these specific parts to cover the whole area. In this study, we demonstrated our first step to stress the importance of the development of the automatic assembly system. As for employing random stimulation commands in our experiment, the primary motivation is to emulate real-world scenarios where the environment is unpredictable, and obstacles or paths cannot always be pre-defined. This approach allows us to assess the robustness and adaptability of the cyborg insects in handling dynamic environments. Moreover, randomized commands prevent the insects from habituating to repetitive stimuli, which might reduce responsiveness over time²². From a performance evaluation perspective, randomization also ensures that the experimental results are not biased by specific control patterns, thus providing a broader understanding of the system's capabilities.

[Changes in manuscript]

The following part was added to the Section 4.6.

Line Number 712-716.

Random electrical stimulation type was chosen to simulate a decentralized, autonomous exploration strategy, which reflects real-world scenarios where multiple agents operate without a pre-defined navigation path. This approach allows for unbiased coverage distribution and reduces dependency on precise localization or predefined control algorithms.

[R3-11] What is the unique function of the proposed deceleration instruction in the arena shown in Fig. 5B(i) that contributes to the coverage improvement? Can specific terrain be designed in the arena to highlight the effect of the deceleration command?

[Reply to this comment]

Thank you for the insightful question. As mentioned in our response to the previous question, in our previous study, we discovered the boosting function of the methyl salicylate on the insects' motion antivenenes level¹². Hence, we used the same chemical stimulation protocol for the coverage task in this paper. However, insects were found being habituated against the repeatedly same stimulation which is discussed in other studies^{11,22}. To effectively avoid such habituation from discounting the covering effect, the deceleration was used here to enable insects keep reactive against the steering stimulation. Specific terrain designs, such as mazes with sharp turns, dead-end paths, or highly confined regions, could further emphasize the effectiveness of the deceleration command. These terrains would simulate real-world scenarios like urban rubble or dense vegetation.

[R3-12] Please give the formula for calculating the coverage rate, as well as the key parameters such as the coverage radius and the trajectory sampling rate when the cockroach is in a certain position. And add the parameter legend in Fig. 5B(iii).

[Reply to this comment]

We included the coverage rate formula and explained key parameters and trajectory sampling rate accordingly. For the legend of Fig. 5B(iii), as the limited space of the figure, we included it in the Fig. 5B(iv), using same colour arrangement for the four insects.

[Changes in manuscript]

The following part was added to the Section 4.6.

Line Number 719-722.

The coverage rate was calculated as the change of covered squares over the change of time, i.e.,

$$\text{Coverage rate} = \text{change of covered squares}/t$$

where t is the change of time. The trajectory sampling rate is 20 Hz.

[R3-13] It is recommended to include control and experimental groups for both non-stimulated and stimulated motion traces, with parameters including distance traveled, head speed, and time spent.

[Reply to this comment]

We included the control group using insect-computer hybrid robots without electrical stimulation and chemical booster. The coverage of the control group achieved 35.3% in 10 minutes (Fig. R3-3), while

the experimental group achieved 75.4% within the same time. Distance travelled, average speed for each insect were also showed in the Table R3-2. Thank you for your suggestions. The comparison between the experimental group and the control group clearly shows the advancements of our stimulation strategy.

[Changes in manuscript]

Table R3-2 was added to Page 17 of Supplementary Materials, as Table. S4.

Fig. R3-3 was added to Page 14 of Supplementary Materials, as Fig. S12.

Table R3-2. Distance travelled and average speed of insects from control and experimental group in 10 minutes.

Group	Insect No	Distance travelled (m)	Average speed (cm/s)
Control group	1	62.9	10.5
	2	65.3	10.9
	3	64.4	10.7
	4	56.3	9.4
Experimental group	1	83.8	13.9
	2	95.8	15.9
	3	108.4	18.1
	4	102.2	17.0

Fig. R3-3. Four insect-computer hybrid robots covering obstructed terrain in 10 minutes with no stimulation applied. The hybrid robots without any stimulation can only move around the releasing point. Their coverage is less than half of the hybrid robots with stimulation.

[R3-14] Please supplement the results of coverage experiments using one, two, and three cockroaches, each conducted at least five times, and report the mean and standard deviation of coverage. Thus, the relationship between the number of cockroach robots and the coverage rate is more convincing.

[Reply to this comment]

Thank you for the suggestions. In the Fig. 5B. iii, the trajectories of the cyborg insects showed even with some trajectories overlapped, there were still some parts only passed by certain insect. For example, Insect 2 passed the right top corner and Insect 1 passed the left wall. Hence, more cyborg insects used for coverage achieved better result than the less insects. Such situations were also found in the previous study¹⁰, indicating this relationship also happened in other terrains.

Nevertheless, we agree with your suggestions that comparison with different numbers of insects is more convincing. Accordingly, we conducted the experiment for different numbers of insects' covering (Fig. R3-4). With the number of the used insects increasing, the average covered area increased.

[Changes in manuscript]

Fig. R3-4 was added to Page 15 of Supplementary Materials, as Fig. S13.

Fig. R3-4. Covered area in one minute with different number of insects. With the number of insects increasing from one to three, the average covered area increased from 0.19 m² to 0.49 m².

[R3-15] The proposed method for direction control through stimulation of the intersegmental membrane in the thorax of cockroaches has not been directly compared with existing antenna-based and abdominal stimulation methods in terms of performance metrics, including the success rate of stimulation, response times, and the physiological impact of long-term stimulation on the cockroaches.

[Reply to this comment]

Thank you for raising this important question. Using the same electrical stimulation parameters, i.e., 0.4 s, 3.0 V, 42 Hz, we controlled the direction of the insects and provided the comparison on the average turning angle, stimulation success rate, and survival rate (1 week after repeatedly 10 times of stimulation) to indicate the physiological impact accordingly. As the stimulation time is fixed as 0.4 s,

the response time of different methods keep same. The insect was considered to have been successfully steered if it rotated according to the direction of the stimulation. From the table, the pronotum-based control performed best in terms of the average turning angle and the stimulation success rate. As summarized in Table R3-3, antennae-based stimulation performs closely to the pronotum-based (11.4% difference in turning angle and 2% difference in success rate). However, abdomen-based stimulation can only achieve less than 1/3 turning angle of the pronotum-based and 24% lower stimulation success. All the insects survived after one week from the 10 times of repeated stimulation.

[Changes in manuscript]

Table R3-3 was added to Page 17 of Supplementary Materials, as Table. S5.

Table R3-3. Comparison of Direction Control.

Stimulation method	Average turning angle	Stimulation success rate	Survival rate
Abdomen-based	20.6°	76%	100%
Antenna-based	66.6°	98%	100%
Pronotum-based	75.2°	100%	100%

[R3-16] How long is the lifespan of the cockroaches implanted using this method compared to those implanted manually by hand. If possible, could you provide additional data on this comparison?

[Reply to this comment]

Insect-computer hybrid robots have similar lifespan with the normal, intact insects. Madagascar hissing cockroaches can live around 2 years under appropriate living conditions²³. So far, we have not observed any situations where lifespan of the insect-computer hybrid robots is shortened due to our experiments. As the time limited, we cannot track their whole life. Nevertheless, for the real use at the post-disaster scenarios, the first 24 hours are very important for the search and rescue²⁴. Considering the rescue team needs time to extricate survivors once they are located by searching robots (cyborg insects) in real search and rescue mission, the time used for searching by cyborg insects is limited to 8 hours. To provide a quantitative understanding of this matter, we compared the locomotion control durability of the insects assembled automatically and manually during the first 8 hours after assembly. Three insects were tested with random steering and deceleration stimulation for each assembly method. For steering stimulation, the insect was considered to have been successfully stimulated if it rotated according to the direction of the stimulation. For deceleration stimulation, successful stimulation was defined as the insect's minimum speed during stimulation being lower than its minimum speed within 0.4 s before stimulation. The results were shown in the Table R3-4. Both the insect-computer hybrid robots prepared automatically and manually were controlled well (over 80% success rate, Table R3-4) throughout the whole 8 hours after assembly. Also, all the insects live well even after one week. Hence, there is no difference on the survival of the cockroaches using different assembly methods. Thank you for your question. We have added the data to the study.

[Changes in manuscript]

Table. R3-4 was added to Page 17 of Supplementary Materials, as Table. S6.

Table R3-4. Locomotion Control Success Rate after assembly finished.

Assembly Method	1 hour	4 hours	8 hours
Automatically assembled	100%	91.7%	91.7%
Manually assembled	100%	83.3%	100%

[R3-17] Section 2.4 reports that assembling four insect-computer hybrid robots took 7 minutes and 48 seconds, whereas Section 2.3.4 indicates that the automatic process requires 68 seconds per insect. This results in a total assembly time for four robots that is approximately 6.88 times longer than for a single robot, suggesting potential scalability issues. It is recommended that the authors provide further clarification and analysis on the time efficiency of assembling multiple hybrid robots consecutively to address this discrepancy.

[Reply to this comment]

Thank you for pointing out this discrepancy. The reported 68 seconds per insect refers strictly to the robotic automatic assembly process, which includes fixing and releasing insect, implanting electrodes and securing the backpack. In contrast, the 7 minutes and 48 seconds required for assembling four robots accounts for additional tasks, such as manually placing the insects on the assembly platform, removing them after assembly, and initializing the robotic system operational program. These preparatory and post-assembly steps are necessary to ensure smooth operation to prepare the multiple cyborg insects. We have clarified this distinction in the revised manuscript to better reflect the realistic workflow and time allocation for consecutive assembly tasks (Fig. R3-2).

[Changes in manuscript]

Fig. R3-2 was added to Page 7 of Supplementary Materials, as Fig. S5.

The following part was added to the Section 2.4.

Line Number 438-447.

The automatic assembly of four insect-computer hybrid robots took 7 min and 48 s. This duration included not only the core assembly process but also additional tasks such as placing the insects on the assembly platform, removing them after assembly, placing the backpacks on the backpack holder, and initializing the robotic system operational program (Fig. S5). These preparatory and post-assembly steps were essential to ensure smooth operation and readiness of multiple insect-computer hybrid robots. The ability to rapidly and efficiently assemble hybrid robots enhances their applicability in time-sensitive missions. To demonstrate the necessity and benefits of scalable production, this study explores terrain coverage in an unknown, obstructed outdoor environment as a fundamental task for multiple-agent systems¹².

3. Concerns regarding the preparation of electrodes:

[R3-18] A subsequent electroless plating process is conducted after multi-material 3D printing to improve its corrosion resistance (Fig. 2B). More data, e.g. comparison on surface morphology by means of optical images or scanning electron microscope, are expected to illustrate the resistance of electrochemical corrosion of the electrode under stimulation cycles with the variation of voltage.

[Reply to this comment]

We have provided a scanning electron microscope (SEM) image to illustrate the surface morphology of the electroless-plated region of the designed bipolar electrode (Fig. R3-5C, D). The electroless plating process enhances the corrosion resistance of the electrode, as reported in previous studies^{25,26}. The electrochemical performance of the electrode may degrade slightly with variations in applied voltage and repeated stimulation cycles. However, our previous experimental observations suggest that the impact is minimal, especially to the locomotion control of the insect (Table R3-4). Thank you for the suggestion.

[Changes in manuscript]

Fig. R3-5 was added to Page 4 of Supplementary Materials, as Fig. S3.

Fig. R3-5. (A) Schematic illustration of the active precursor copper plating process on the resin structure. The surface resistivity is measured to be between $2\Omega/\square$ and $8.5\Omega/\square$, indicating the successful formation of a conductive copper layer. The inset shows the surface morphology of the resin structure with a scale bar of 1 mm. (B) Phase angle vs. frequency. (C) SEM image of the resin structure prior to plating, displaying a relatively smooth surface with oriented microstructures, and the inset shows the sampling location within the resin structure (scale bar = 1 mm). (D) Characterization of the plated copper layer: (i) SEM image of the plated surface, showing a uniform granular structure with

well-defined particle size distribution (scale bar = 10 μm); (ii) EDS elemental mapping of Cu, confirming the homogeneous distribution of copper over the plated surface; and (iii) EDS spectrum, identifying prominent Cu peaks, along with minor Pd and O peaks, indicating the successful catalytic activity of Pd and the formation of a stable copper layer.

[R3-19] Fig. 2F shows the impedance characteristics of bipolar electrodes, which was stably below 70 Ω . It is recommended to add detailed steps and instruments used in the impedance test. Besides, the corresponding phase changes with frequency are supposed to be provided to illustrate the comprehensive electrochemical properties of electrodes.

[Reply to this comment]

For the impedance test in the Fig. 2, we used Auto-Balancing Bridge Method to accurately get the impedance value of the bipolar electrodes. The instruments used for this experiment is Hioki Im3570 Impedance Analyzer. Surface resistance was measured to be between 2 Ω/\square and 8.5 Ω/\square using Four-probe Method with MCP-T700 Resistivity Meters (Fig. R3-5A). Also, we provide phase changes with frequency accordingly using ET4410 LCR Meter (Fig. R3-5B). Thanks to your suggestions, the properties of electrodes are more comprehensive now.

[Changes in manuscript]

Fig. R3-5 was added to Page 5 of Supplementary Materials, as Fig. S3.

The following part was added to the Section 4.2.1.

Line Number 535-537.

After the electroless plating, impedance of bipolar electrodes was tested using Auto-Balancing Bridge Method and shown in the Fig. 2F. The instrument to conduct this experiment was Hioki Im3570 Impedance Analyzer.

[R3-20] In Table 4, the material parameters are defined for the finite element analysis of implantation stress. It would be better to provide the element type for the FEA model.

[Reply to this comment]

The element type we used for the FEA model in Ansys was PLANE182. We have provided this information in the manuscript accordingly.

[Changes in manuscript]

The following part was added to the Section 4.2.1.

Line Number 544-545.

The finite element analysis for membrane damage simulation was conducted using Lsdyna part in Ansys 19.0 with element type of PLANE182.

4. Some opinions on writing and pictures:

[R3-21] The primary focus of this paper is the cockroach control paradigm and the implantation of an automatic robotic arm. Therefore, could these two components be presented in Section II instead of the content related to electrodes?

[Reply to this comment]

We agree that the primary focus of this study is the cockroach control paradigm and the automatic robotic arm for implantation. However, the bipolar electrodes play a fundamental role in both aspects and are essential for achieving effective electrical stimulation and successful automatic assembly. Therefore, discussing the electrode design in Section II is critical to ensuring a clear understanding of the entire system.

[R3-22] There are some images that remain too blurry, even in the largest and highest definition versions. For example, the legend on the left side of each figure in Fig. 2D is unclear. Additionally, Fig. 3B is also blurry.

[Reply to this comment]

We provided better images of Fig. 2D and Fig. 3B for the readers accordingly. For Fig. 3B, we used blue lines to indicate the motion of the two forelegs. Also, in the Movie S2, we showed the real reaction of the insect. For Fig. 2D, as the limited space for the figure arrangement, we included clearer image inside the Supplementary Materials as Fig. S4.

[Changes in manuscript]

Fig. S4 in Page 6 of Supplementary Materials was used to show the legend of Fig. 2D.

Fig. 3B. was replaced by a clearer image and the forelegs was indicated with blue lines.

[R3-23] The reviewer commented on the language of the manuscript, noting that it contains grammatical and spelling errors that impact its clarity and readability. These linguistic issues must be addressed.

[Reply to this comment]

Excuse us the original manuscript had errors in English. We have brushed up the manuscript carefully, and also, we have ordered a professional language editing service from LetPub which has corrected the grammatical and spelling errors accordingly throughout the whole manuscript (certificate number: PR_250313M37).

[R3-24] The organization and logic of the manuscript require careful refinement. The abstract centers on the automatic assembly system for insect robots developed in this study, while the introduction highlights the discovery of directional control sites based on the stimulation of the intersegmental membrane between the pronotum and the mesothorax, with only a brief mention of the assembly system at the end. This structure may lead readers to believe that the authors have not clearly articulated the contributions and logical progression of the paper.

[Reply to this comment]

Thank you very much for pointing out the organization issue. We modified the abstract parts to make the content more clearly to the readers.

[Changes in manuscript]

The following part was added to the Abstract.

Line Number 35-55

In this study, we identified the intersegmental membrane between the pronotum and mesothorax of the Madagascar hissing cockroach as an effective stimulation site for directional and speed control. To stably transfer the electrical stimulation to the intersegmental membrane, a pair of bipolar electrodes was designed specifically for the cockroach. To enable precise and scalable implantation of the bipolar electrodes, we developed an automatic assembly system that integrates a robotic arm for bipolar electrode implantation, a deep learning-based vision system for site identification, and a dedicated fixation structure for the insect. This system achieved automatic assembly process in 68 seconds. The automatically assembled hybrid robots demonstrated steering control (over 70 ° for 0.4 s stimulation) and deceleration control (68.2% speed reduction for 0.4 s stimulation), achieving comparable locomotion control performance to manually assembled systems (Student's t-test: $P = 0.62$ for left turns, $P = 0.50$ for right turns, $P = 0.21$ for deceleration). To demonstrate the control effect, a hybrid robot was controlled to follow an S-shape line. The path taken by the hybrid robot highly overlaps with the predefined line. Furthermore, a multi-agent system of four hybrid robots successfully covered 80.25% of an obstructed outdoor terrain within 10 minutes and 31 seconds, demonstrating the feasibility of mass production for practical deployment. This work establishes a scalable strategy for automating the fabrication of insect-computer hybrid robots, ensuring efficient and reproducible assembly process while maintaining robust locomotion control.

5. About the deep learning-based Detection of the Pronotum part:

[R3-25] In the section on the Detection of Pronotum using deep learning, the paper evaluates segmentation models such as UNet, Deeplabv3, TransUNet, and Segment Anything, all of which are methods from or before 2023. Given the emergence of new models and architectures in the field after SAM, these should also be assessed.

[Reply to this comment]

We have identified new models that were introduced in 2024 and 2025, (LM-Net²⁸, InceptionNeXt²⁹, EMCAD³⁰, SHViT³¹ and SAM2³²) for comparison. Based on the performance metrics of these models, we conclude that these new models cannot significantly increase the accuracy (Fig. R3-6, Table R3-5). Hence, we continued using our previous model and compared with different loss function (Fig. R3-7, Table R3-6).

[Changes in manuscript]

Table R3-5 was added to Page 47 of Manuscript, as Table 1.

Table R3-6 was added to Page 47 of Manuscript, as Table 2.

Fig. R3-6 was added to Page 8 of Supplementary Materials, as Fig. S6.

Fig. R3-7 was added to Page 9 of Supplementary Materials, as Fig. S7.

The following part was added to the Section 2.3.2.

Line Number 290-296.

An evaluation was conducted on several widely used segmentation models, including UNet³³, Deeplabv3³⁴, TransUNet³⁵, as well as recently developed segmentation models, such as Segment Anything³⁶, Segment Anything2³², LM-Net²⁸, InceptionNeXt²⁹, EMCAD³⁰ and SHViT³¹, for their accuracy in pronotum segmentation. The models' performance was measured by the mean intersection over union (mIoU) score, mean Dice similarity coefficient (mDSC), and mean squared error (MSE) of p_R prediction (Table 1).

Line Number 300-308

The other hybrid models, LM-Net, SHViT and EMCAD were designed to be lightweight models geared towards faster inference times, causing poor performance, apart from LM-Net, which had achieved comparable scores with the other models despite having significantly less parameters. The CNN-based models UNet, Deeplabv3 and InceptionNeXt managed to achieve better segmentation accuracies compared to the foundational ViT-based models SAM and SAM2, with Deeplabv3 even having similar performance to TransUNet in all three metrics as it was generally able to segment out most of the pronotum but did not precisely identify the lower border of the pronotum, causing slightly poorer scores on the three metrics.

Line Number 311-319

The DSC and BCE loss functions generally outperformed the bDoU loss function on the mIoU and mDSC metrics, as they evaluated the prediction of the full pronotum while the bDoU loss focused primarily on the boundaries of the segmented objects. However, the bDoU loss function achieved the best MSE score among the three, highlighting its effectiveness in training the model to learn object boundaries. Ultimately, the chosen deep learning solution was the TransUNet model trained with the BCE loss

function (Fig. 4C), as it had the best mIoU and mDSC scores and only slightly underperformed on the MSE metric compared to the model trained with bDoU loss.

The following part was added to the Section 4.3.2.

Line Number 600-605.

Hence, a mixture of CNN-based, ViT-based and CNN-Transformer hybrid models were surveyed for our application, to evaluate the performance of different variations of vision model architectures. The following models were trained and evaluated on our dataset of cockroaches: UNet³³, TransUNet³⁵, Deeplabv3³⁴ and Segment Anything³⁶, Segment Anything2³², LM-Net²⁸, InceptionNeXt²⁹, EMCAD³⁰ and SHViT³¹.

Line Number 606-622.

The UNet, Deeplabv3 and InceptionNeXt were CNN-based models, with the Deeplabv3 model using the ResNet-101³⁷ and the InceptionNeXt model using ResNet-50³⁷, which were both pretrained on ImageNet³⁸, as their backbone. The TransUNet LM-Net, EMCAD and SHViT models were hybrid models, having a CNN-Transformer hybrid encoder, to combine the strengths of CNNs in locality and ViTs in the global context. The encoder for the TransUNet model combines ResNet-50³⁷ and ViT-B³⁹, which was pretrained on ImageNet³⁸. The EMCAD model uses the Pyramid ViT v2 (PVTv2)⁴⁰ as its encoder, which was pretrained on ImageNet³⁸. Segment Anything and Segment Anything2 were promptable foundational models, with the former using a ViT-H³⁹ model and the latter using a Hiera-L⁴¹ model, and box prompts were used to specify the pronotum as the segmentation target. Apart from Segment Anything and Segment Anything2, the other models were trained on the cockroach dataset, with a batch size of 32, a two-phase learning rate scheduler, linearly raising the learning rate from 0.0001 to 0.001 during the 10 epoch warmup phase and subsequently cosine annealing was used to decay the learning after the warmup phase, Adaptive Moment Estimation as the optimizer and Binary Cross Entropy (BCE) as the loss function for 300 epochs with no early stopping condition.

Table R3-5. Comparison of models on the cockroach test samples dataset

Model	Params (M)	mIoU	mDSC	MSE (ρ_R)
LM-Net	5	0.8969	0.9448	1.970
SHViT	18	0.8186	0.8990	3.722
EMCAD	26	0.7688	0.8689	3.184
UNet	31	0.8741	0.9318	2.382
Deeplabv3	61	0.9245	0.9605	1.786
TransUNet	105	0.9326	0.9650	1.749
InceptionNeXt	193	0.8567	0.9222	2.603
Segment Anything2	224	0.8040	0.8885	3.246
Segment Anything	636	0.8471	0.9161	2.387

Table R3-6. Comparison of loss functions used for TransUNet training

Loss Function	mIoU	mDSC	MSE (p_R)
BCE Loss	0.9326	0.9650	1.749
DSC Loss	0.9289	0.9630	1.807
bDoU Loss	0.9283	0.9626	1.617

Fig. R3-6. Qualitative comparison of segmentation models on the cockroach dataset.

Fig. R3-7. Qualitative comparison of TransUNet when trained on different loss functions: BCE Loss, DSC Loss and bDoU Loss.

[R3-26] The amount of training and testing data in the Detection of Pronotum is unclear; the authors mention experiments with 29 unique cockroaches but do not specify the number of images in the training and testing sets.

[Reply to this comment]

Thank you for pointing out the unclear illustration. Twenty original images were used for the testing set, while the other original images were heavily augmented to produce 6570 images for the training set. The training set was further split 80/20 for the train/validation split, leading to 5254 images for training and 1316 images for validation.

[Changes in manuscript]

Table R3-7 was added to Page 47 of Manuscript, as Table 3.

The following part was added to the Section 4.3.2.

Line Number 624-637.

The robotic arm was placed in a fixed position to capture images with the Intel RealSense D435 camera. We used 256 × 256-pixel crop from the original image, centered around the pronotum as the input for training the models. This approach shortened the training and inference times. Of the collected images, 20 were used as test cases for model evaluation, while the remaining were used for training. Training data were augmented to enhance model robustness against variances in pronotum rotation and shape. Asymmetrical scaling of the training samples in the x- and y- axes, using scaling values between 0.8 to 1.2, was applied to the training samples using bilinear interpolation to generate more unique pronotum shapes to simulate the varying pronotum sizes of cockroaches. Subsequently, rotations between – 6 ° and 6 ° were applied to accommodate inconsistencies in positions when cockroaches were mounted on the 3D structure. The final training dataset contained 6570 images after data augmentation, and furthermore an 80/20 train/validation split was used, leading to 5254 images used for training and 1316 images used for validation.

Table R3-7. Comparison of TransUNet models trained on datasets with various levels of data ablation

Training Image Data			Metrics		
Original	Asymmetrically scaled	Rotated	mIoU	mDSC	MSE
✓			0.8721	0.9296	2.326
✓	✓		0.8847	0.9347	2.090
✓		✓	0.9211	0.9588	2.152
✓	✓	✓	0.9326	0.9650	1.749

[R3-27] Additionally, the paper does not detail the quantity of data after augmentation or quantify the impact of using and not using data augmentation on the performance of the segmentation models. [Reply to this comment]

Thank you for kind suggestions. We mainly employed 2 types of data augmentation to generate artificial data:

1. Asymmetrical scaling in the x and y axes, to generate unique pronotum shapes, allowing the model to be more robust to different pronotum shapes
2. Image rotation, to allow the model to be more robust in the inconsistencies of the cockroach orientations when mounted on the 3D structure.

We compared the accuracy of the model (Table R3-7, Fig. R3-8) when:

- a) Trained with original data and augmented data using 1 and 2
- b) Trained with original data and augmented data only using 1
- c) Trained with original data and augmented data only using 2

[Changes in manuscript]

Table R3-7 was added to Page 47 of Manuscript, as Table 3.

Fig. R3-8 was added to Page 10 of Supplementary Materials, as Fig. S8.

The following part was added to the Section 2.3.2.

Line Number 320-329.

Data ablation study was also conducted to evaluate how the effect of different data augmentation methods on the training images impacts the segmentation performance. Performance based on mIoU, mDSC and MSE of p_R prediction, is shown in Table 3. The TransUNet model had achieved significantly better scores across all three metrics when trained with augmented data, compared to when trained with only original data. The model trained on the asymmetrically scaled augmented data had shown to have slightly better performance in segmenting the boundaries compared to the model trained on the rotated augmented data, attributed to the robustness to variations in the shapes of pronotums. However, the latter had a better segmentation accuracy, as indicated by the larger mIoU and mDSC scores.

The following part was added to the Section 4.3.2.

Line Number 638-642.

A data ablation study was conducted, where the performance of the models was compared when trained on four conditions: only original unaugmented images (9 images), original images with augmentation using asymmetrical scaling in x-y axes (1316 images), original images with augmentation using rotation (90 images) and the full augmented training dataset (6570 images).

Fig. R3-8. Qualitative comparison of TransUNet when trained on different data mixtures. (a) original unaugmented images, (b) asymmetrically scaled images, (c) rotated images.

=====

Overall, all the authors deeply appreciate the valuable comments and suggestions provided by the reviewers and the editor. Thanks to their insightful feedback, we have identified the shortcomings of the original manuscript and have added additional experimental data and expanded the discussion in the revised version. We believe that the revised manuscript is significantly more convincing than the original.

Again, thank you for reading this letter through to the end.

We have learned a great deal from their input and are committed to applying the developed technology in practical fields to help save lives.

Sincerely,

All the authors represented by
 Hirotaka Sato, Professor
 Nanyang Technological University, Singapore
 hirosato@ntu.edu.sg / +65 6790 5010

References:

1. Lin Q, Li R, Zhang F, Kai K, Ong ZC, Chen X, et al. Resilient conductive membrane synthesized by in-situ polymerisation for wearable non-invasive electronics on moving appendages of cyborg insect. *Npj Flex Electron*. 2023 Sep 1;7(1):1–10.
2. Latif T, Bozkurt A. Line following terrestrial insect biobots. In: 2012 Annual International Conference of the IEEE Engineering in Medicine and Biology Society. 2012. p. 972–5.
3. Nguyen HD, Dung VT, Sato H, Vo-Doan TT. Efficient autonomous navigation for terrestrial insect-machine hybrid systems. *Sens Actuators B Chem*. 2023 Feb 1;376:132988.
4. Vo Doan TT, Tan MYW, Bui XH, Sato H. An Ultralightweight and Living Legged Robot. *Soft Robot*. 2018 Feb;5(1):17–23.
5. Li R, Lin Q, Tran-Ngoc PT, Le DL, Sato H. Smart insect-computer hybrid robots empowered with enhanced obstacle avoidance capabilities using onboard monocular camera. *Npj Robot*. 2024 Jun 24;2(1):1–10.
6. Erickson JC, Herrera M, Bustamante M, Shingiro A, Bowen T. Effective Stimulus Parameters for Directed Locomotion in Madagascar Hissing Cockroach Biobot. *PLOS ONE*. 2015 Aug 26;10(8):e0134348.
7. Titlow JS, Majeed ZR, Hartman HB, Burns E, Cooper RL. Neural Circuit Recording from an Intact Cockroach Nervous System. *J Vis Exp JoVE*. 2013 Nov 4;(81):50584.
8. Barriga-Rivera A, Guo T, Yang CY, Abed AA, Dokos S, Lovell NH, et al. High-amplitude electrical stimulation can reduce elicited neuronal activity in visual prosthesis. *Sci Rep*. 2017 Feb 17;7(1):42682.
9. Wu GK, Ardeshipour Y, Mastracchio C, Kent J, Caiola M, Ye M. Amplitude- and frequency-dependent activation of layer II/III neurons by intracortical microstimulation. *iScience*. 2023 Oct 6;26(11):108140.
10. Lin Q, Kai K, Nguyen HD, Sato H. A newly developed chemical locomotory booster for cyborg insect to sustain its activity and to enhance covering performance. *Sens Actuators B Chem*. 2024 Jan;399:134774.
11. Bernal-Gamboa R, García-Salazar J, Gámez AM. Analysis of Habituation Learning in Mealworm Pupae (*Tenebrio molitor*). *Front Psychol [Internet]*. 2021 [cited 2022 Jun 1];12. Available from: <https://www.frontiersin.org/article/10.3389/fpsyg.2021.745866>
12. Lin Q, Kai K, Nguyen HD, Sato H. A newly developed chemical locomotory booster for cyborg insect to sustain its activity and to enhance covering performance. *Sens Actuators B Chem*. 2024 Jan 15;399:134774.
13. Lin Q, Li R, Zhang F, Kai K, Ong ZC, Chen X, et al. Resilient conductive membrane synthesized by in-situ polymerisation for wearable non-invasive electronics on moving appendages of cyborg insect. *Npj Flex Electron*. 2023 Sep 1;7(1):1–10.
14. Latif T, Whitmire E, Novak T, Bozkurt A. Sound Localization Sensors for Search and Rescue Biobots. *IEEE Sens J*. 2016 May;16(10):3444–53.

15. Ariyanto M, Refat CMM, Yamamoto K, Morishima K. Feedback control of automatic navigation for cyborg cockroach without external motion capture system. *Heliyon* [Internet]. 2024 Mar 15 [cited 2025 Jan 17];10(5). Available from: [https://www.cell.com/heliyon/abstract/S2405-8440\(24\)03018-4](https://www.cell.com/heliyon/abstract/S2405-8440(24)03018-4)
16. Tran-Ngoc PT, Le DL, Chong BS, Nguyen HD, Dung VT, Cao F, et al. Intelligent Insect–Computer Hybrid Robot: Installing Innate Obstacle Negotiation and Onboard Human Detection onto Cyborg Insect. *Adv Intell Syst.* 2023;5(5):2200319.
17. Ma S, Chen Y, Yang S, Liu S, Tang L, Li B, et al. The Autonomous Pipeline Navigation of a Cockroach Bio-Robot with Enhanced Walking Stimuli. *Cyborg Bionic Syst.* 2023 Nov 8;4:0067.
18. Liu Z, Gu Y, Yu L, Yang X, Ma Z, Zhao J, et al. Locomotion Control of Cyborg Insects by Charge-Balanced Biphasic Electrical Stimulation. *Cyborg Bionic Syst.* 2024 Jul 5;5:0134.
19. Bai Y, Tran Ngoc PT, Nguyen HD, Le DL, Ha QH, Kai K, et al. Swarm navigation of cyborg-insects in unknown obstructed soft terrain. *Nat Commun.* 2025 Jan 6;16(1):221.
20. Fonseca N, Thummalapalli SV, Jambhulkar S, Ravichandran D, Zhu Y, Patil D, et al. 3D Printing-Enabled Design and Manufacturing Strategies for Batteries: A Review. *Small.* 2023;19(50):2302718.
21. Wang D, Dong Y, Lian J, Gu D. Adaptive end-effector pose control for tomato harvesting robots. *J Field Robot.* 2023;40(3):535–51.
22. Li R, Lin Q, Kai K, Nguyen HD, Sato H. A Navigation Algorithm to Enable Sustainable Control of Insect-Computer Hybrid Robot with Stimulus Signal Regulator and Habituation-Breaking Function. *Soft Robot* [Internet]. 2023 Dec 29 [cited 2024 Jan 31]; Available from: <https://www.liebertpub.com/doi/10.1089/soro.2023.0064>
23. Mulder P, Shufran A. Madagascar Hissing Cockroaches: Information and Care.
24. Hakami A, Kumar A, Shim SJ, Nahleh YA. Application of Soft Systems Methodology in Solving Disaster Emergency Logistics Problems. 2013;7(12).
25. Sankara Narayanan TSN, Baskaran I, Krishnaveni K, Parthiban S. Deposition of electroless Ni–P graded coatings and evaluation of their corrosion resistance. *Surf Coat Technol.* 2006 Mar 15;200(11):3438–45.
26. Xie J, Wang Z, Bai X, Li H, Wang S, Hao W, et al. Recycled industrial waste silicon steel as high-performance electrode for oxygen evolution reaction using electroless plating surface modification. *Appl Surf Sci.* 2025 Feb 15;682:161747.
27. Paivana G, Barmpakos D, Mavrikou S, Kallergis A, Tsakiridis O, Kaltsas G, et al. Evaluation of Cancer Cell Lines by Four-Point Probe Technique, by Impedance Measurements in Various Frequencies. *Biosensors.* 2021 Sep 18;11(9):345.
28. Lu Z, She C, Wang W, Huang Q. LM-Net: A light-weight and multi-scale network for medical image segmentation. *Comput Biol Med.* 2024 Jan 1;168:107717.
29. Yu W, Zhou P, Yan S, Wang X. InceptionNeXt: When Inception Meets ConvNeXt. In 2024 [cited 2025 Mar 15]. p. 5672–83. Available from: https://openaccess.thecvf.com/content/CVPR2024/html/Yu_InceptionNeXt_When_Inception_Meets_ConvNeXt_CVPR_2024_paper.html

30. Rahman MM, Munir M, Marculescu R. EMCAD: Efficient Multi-scale Convolutional Attention Decoding for Medical Image Segmentation. In 2024 [cited 2025 Mar 15]. p. 11769–79. Available from: https://openaccess.thecvf.com/content/CVPR2024/html/Rahman_EMCAD_Efficient_Multi-scale_Convolutional_Attention_Decoding_for_Medical_Image_Segmentation_CVPR_2024_paper.html
31. Yun S, Ro Y. SHViT: Single-Head Vision Transformer with Memory Efficient Macro Design. In 2024 [cited 2025 Mar 15]. p. 5756–67. Available from: https://openaccess.thecvf.com/content/CVPR2024/html/Yun_SHViT_Single-Head_Vision_Transformer_with_Memory_Efficient_Macro_Design_CVPR_2024_paper.html
32. Ravi N, Gabeur V, Hu YT, Hu R, Ryali C, Ma T, et al. SAM 2: Segment Anything in Images and Videos [Internet]. arXiv; 2024 [cited 2025 Mar 15]. Available from: <http://arxiv.org/abs/2408.00714>
33. Ronneberger O, Fischer P, Brox T. U-Net: Convolutional Networks for Biomedical Image Segmentation [Internet]. arXiv; 2015 [cited 2024 Aug 28]. Available from: <http://arxiv.org/abs/1505.04597>
34. Chen LC, Papandreou G, Schroff F, Adam H. Rethinking Atrous Convolution for Semantic Image Segmentation [Internet]. arXiv; 2017 [cited 2024 Aug 28]. Available from: <http://arxiv.org/abs/1706.05587>
35. Chen J, Lu Y, Yu Q, Luo X, Adeli E, Wang Y, et al. TransUNet: Transformers Make Strong Encoders for Medical Image Segmentation [Internet]. arXiv; 2021 [cited 2024 Aug 28]. Available from: <http://arxiv.org/abs/2102.04306>
36. Kirillov A, Mintun E, Ravi N, Mao H, Rolland C, Gustafson L, et al. Segment Anything [Internet]. arXiv; 2023 [cited 2024 Aug 28]. Available from: <http://arxiv.org/abs/2304.02643>
37. He K, Zhang X, Ren S, Sun J. Deep Residual Learning for Image Recognition [Internet]. arXiv; 2015 [cited 2024 Aug 28]. Available from: <http://arxiv.org/abs/1512.03385>
38. Deng J, Dong W, Socher R, Li LJ, Li K, Fei-Fei L. ImageNet: A large-scale hierarchical image database. In: 2009 IEEE Conference on Computer Vision and Pattern Recognition [Internet]. 2009 [cited 2024 Aug 28]. p. 248–55. Available from: <https://ieeexplore.ieee.org/abstract/document/5206848>
39. Dosovitskiy A, Beyer L, Kolesnikov A, Weissenborn D, Zhai X, Unterthiner T, et al. An Image is Worth 16x16 Words: Transformers for Image Recognition at Scale [Internet]. arXiv; 2021 [cited 2024 Aug 28]. Available from: <http://arxiv.org/abs/2010.11929>
40. Wang W, Xie E, Li X, Fan DP, Song K, Liang D, et al. PVT v2: Improved baselines with Pyramid Vision Transformer. *Comput Vis Media*. 2022 Sep 1;8(3):415–24.
41. Ryali C, Hu YT, Bolya D, Wei C, Fan H, Huang PY, et al. Hiera: A Hierarchical Vision Transformer without the Bells-and-Whistles. In: Proceedings of the 40th International Conference on Machine Learning [Internet]. PMLR; 2023 [cited 2025 Mar 15]. p. 29441–54. Available from: <https://proceedings.mlr.press/v202/ryali23a.html>

**Reviewer #1:**

**[R1-1]** The authors addressed the reviewers' comments by conducting
additional experiments, providing clarifications, and revising the manuscript.
This included clarifying locomotion control in the abstract, detailing the manual
assembly process and challenges, providing explanations and citations for
observed neural activity, correcting misleading statements about deceleration
control, explaining the vertical backpack attachment, addressing hook failure
issues, and clarifying the role of methyl salicylate in the covering mission. These
changes improved the manuscript's clarity, completeness, and persuasiveness.

There are some minor suggestions for improvements:

[Reply to this comment]

We appreciate your guidance throughout the review process. In this round, your
specific suggestions regarding deceleration control mechanisms, sensor
integration details, and figure improvements have been especially helpful.
Building on your previous recommendations about locomotion control
clarification and methodological transparency, collectively enhancing both the
technical rigor and presentation quality of our work.

**[R1-2]** If possible, the authors could provide a more detailed discussion on how
deceleration control complements steering in obstacle avoidance. A
comparative discussion or comments with existing methods (e.g., forward
motion + steering or deceleration motion + steering) would strengthen the
author's argument.

[Reply to this comment]

Following your suggestion, we have additionally discussed how deceleration
control complements steering in obstacle avoidance. Specifically, the
deceleration reduces the forward momentum of the hybrid robots, allowing
more time and spatial flexibility for steering commands to take effect. This
mechanism is particularly important when navigating complex or cluttered
environments. Recent biological studies have demonstrated similar strategies:
for example, *Holcocephala fusca* uses a combined guidance strategy,
integrating both obstacle avoidance and target pursuit, where speed reduction
enhances steering performance when facing obstacles ¹. Similarly, studies on
bumblebees show that when flying in cluttered environments, insects actively
reduce their speed based on visual flow cues from nearby obstacles to minimize
collision risks ².

These findings suggest that in confined or obstructed terrains, the combination
of deceleration and steering provides a more effective navigation strategy.

Therefore, introducing deceleration control in our insect-computer hybrid robots
helps improve their ability to negotiate obstacles.

**[R1-3]** - Section 2.4 describes the locomotion control experiments. While the
authors have added a section on the S-shape line following, it lacks details
about the control algorithm or strategy used. Including this information would
enhance the reproducibility of the study.

[Reply to this comment]

Thank you for pointing out the control method used for S-shape line following.
In this experiment, we adopted manual control for navigation of the cyborg
insect, consistent with previous studies in literature. Specifically, an operator
triggered stimulation signals based on insect's alignment with the predefined
S-shape path³⁻⁷.

[Changes in manuscript]

The following part was added to the Section 2.4.

*Line Number 435 – 436.*

*Besides, an insect-computer hybrid robot was demonstrated to follow an S-*
*shape line via an operator's command (Movie S5).*

**[R1-4]** - The manuscript mentions potential sensor integration but lacks
specifics. The authors could outline feasible sensor types (e.g., gas sensors,
thermal array for search-and-rescue, etc) and their impact on payload capacity.

[Reply to this comment]

We appreciate the reviewer's insightful comment regarding the need for
specificity in sensor integration. In our study, we utilized the Madagascar
hissing cockroach (*Gromphadorhina portentosa*) as the live creature for our
*insect-computer hybrid robots*. This species can carry additional payloads up to
approximately 15g⁸. This allows for the integration of lightweight sensors
essential for search and rescue (SAR) operations, including infrared (IR)
cameras, RGB cameras, and microphones for human detection and the Inertial
measurement units (IMU) for localization, which are:

- • Infrared (IR) Camera for Thermal Detection: A low-resolution thermopile
array sensor can be used to detect human body heat signatures in low-
visibility conditions, especially inside the collapsed building. For instance,

a 32×32 pixels IR camera with a 90°×90° field of view (~ 0.9 g weight),
has been successfully integrated into cyborg insect, enabling human
detection within a range of 0.5 to 1.5 meters⁹. For enhanced detection
range, we can use the better IR camera with higher resolution. For
example, the FLIR Lepton 3.5 thermal camera module provides a
160×120 pixels resolution and weighs approximately 0.9 g. Its compact
size (10.50×12.70×7.14 mm³) and low power consumption (around 16
109 mW with 10% duty cycle) make it suitable for implementation on cyborg
insect¹⁰.

- • RGB Camera for Visual Recognition: In environments with sufficient
lighting, RGB camera can provide visual of survivors and assist in
environmental mapping. For instance, the OV2640 camera (320×240
pixels resolution, consumes 125 mW at 15fps, 4.7 g weight), has been
effectively used in insect-computer hybrid robots for navigation and
obstacle avoidance tasks¹¹. In addition, an LED light source can be
integrated to assist in human detection under low-light conditions¹².
- • Microphone Array for Acoustic Detection: In case the human is trapped
in the rubble and the camera sensors cannot capture the heat or shape
of a partial body, the microphone can be used to find human presence
through voice or tapping sounds. The detection and localisation of sound
sources has been demonstrated through the use of single and tri-
directional microphone arrays (with weights of 1.6 g and 3.4 g, and
power consumption of 36 mW and 63 mW, respectively) mounted on
cyborg cockroaches in research⁸. These systems can detect sound
sources up to 75 cm away.
- • Inertial measurement units (IMU): In outdoor environments where
external monitoring systems (e.g., the VICON optical motion capture)
are unavailable, onboard localization becomes crucial for autonomous
navigation. A low-power MEMS IMU, such as MPU9250 combines a 3-
axis gyroscope, accelerometer, and magnetometer in a compact 3 × 3 ×
1 mm³ package (~ 0.11 g weight) and consuming around 3.7 mA at
3.3V¹³. Its integration enables the estimation of the insect's position by
analyzing body vibrations during locomotion^{9,14}.

Besides, human presence can also be indicated by volatile organic compounds
(VOCs) like ammonia and acetone emitted through breath and sweat^{15,16}.
Ammonia concentrations typically range from 0.5 to 2.0 ppm, while acetone
levels vary between 0.2 and 0.9 ppm in healthy individuals¹⁷. In field trials,
mobile robots that are equipped with gas sensors, including metal-oxide-
semiconductor (MOS) sensors and photoionization detectors (PIDs), have
effectively detected these VOCs, thereby assisting in the identification of
individuals hiding beneath rubble^{18–20}.

However, integrating gas sensors into cyborg insect platforms presents notable
challenges:

- • Size, Weight, and Power Constraints: Even compact sensors like the
Senseair S8 CO₂ sensor (33.9 × 19.8 × 8.7 mm³, 5 g, power
consumption ~81 mW)²¹ or the SCD30 sensor (35 × 23 × 7 mm³, 3.4 g,
~62.7 mW at 2-second intervals)²² may be too large or heavy for cyborg
insects, potentially hindering mobility. The DFRobot SEN0567 NH₃
sensor has a smaller size than the CO₂ sensor (13 × 13 × 2.5 mm³) but
it is still heavy and consumes higher power (~450 mW)²³.
- • Motion-Induced Instability and Environmental Sensitivity: The
movements and orientation changes of *insect-computer hybrid robots*
can lead to inconsistent airflow over the sensors. Thus, it will make
unstable readings for both CO₂ and NH₃ sensors. Additionally, NH₃
sensors are particularly sensitive to environmental factors like humidity
and temperature fluctuations ²⁴, which can further affect their
performance in the variable conditions typically encountered in disaster
environments.

[Changes in manuscript]

We have added the summary of the above-mentioned potential sensor
integration to the last paragraph (page 20) of Results and Discussion.

*Line Number 474 – 479.*

*To enhance their functionality, lightweight miniaturized thermal and RGB*
*cameras, microphones, and IMUs can be integrated for human detection and*
*localization, though gas sensor integration remains technically challenging due*
*to size, power, and environmental constraints. Altogether, this work establishes*
*a foundational platform for scalable manufacturing and real-world deployment*
*of cyborg insects in complex, unstructured environments.*

**[R1-5]** - Some figures, like Fig. 2D, are noted to be blurry. High-resolution
versions of all images should be used for clarity.

[Reply to this comment]

We have revised the arrangement of Fig. 2D to make it clearer. Thanks to your
suggestion, the overall readability of the paper has been improved.

*Fig. R1. Finite element modeling and analysis during the process of bipolar*
 *electrodes implantation into the intersegmental membrane between the*
 *cockroach's pronotum and mesothorax.*

[Changes in manuscript]

*We have replaced Fig. 2D with a higher-resolution and better-arranged version,*
 *now presented as Fig. R1.*

**[R1-6]** - Consider adding scale bars to all relevant microscopy images to
 provide a clear size context.

[Reply to this comment]

*We have added scale bars to the relevant microscopy images accordingly.*

*Fig. R2. (A) SEM image of the resin structure prior to plating, displaying a*
 *relatively smooth surface with oriented microstructures, and the inset shows the*
 *sampling location within the resin structure (scale bar = 1 mm). (B)*

*Characterization of the plated copper layer: (i) SEM image of the plated surface,*
*showing a uniform granular structure with well-defined particle size distribution*
*(scale bar = 10 μm); (ii) EDS elemental mapping of Cu, confirming the*
*homogeneous distribution of copper over the plated surface; and (iii) EDS*
*spectrum, identifying prominent Cu peaks, along with minor Pd and O peaks,*
*indicating the successful catalytic activity of Pd and the formation of a stable*
*copper layer.*

[Changes in manuscript]

*Fig. R2 has been added to replace Fig. S3C and Fig. S3D.*

**[R1-7]** - Fig. 5B(iii): Please add a color legend for the trajectory plot and indicate
the initial position(s) of the cyborg insects. Same as Fig. S12.

[Reply to this comment]

Yes, that's much better. We have provided color legends to indicate the cyborg
insects and their initial positions accordingly. Thank you for your suggestion.

*Fig. R3. (A) Trajectories of the four insect-computer hybrid robots with electrical*
*stimulation and chemical boosting during the mission. (B) Four insect-computer*
*hybrid robots covering obstructed terrain in 10 minutes with no stimulation*
*applied.*

[Changes in manuscript]

*Fig. R3A has replaced Fig. 5B(i) with the updated legend and position indicators.*
*Fig. R3B has been added to replace the original Fig. S11 for consistency.*

**Reviewer #2:**

**[R2-1]** In the second review, I think the authors have addressed most of my
concerns. The manuscript has become more convincing with newly added
details and experiments. The proposed automatic assembly strategy is an
innovative work on insect-computer hybrid robots, which will reduce the mass-
producing time and cost. The novelty and completeness meet the requirement
of the journal. Thus, I think this manuscript is suitable for publication on Nature
Communications after making minor revisions. Please follow the comments
below.

[Reply to this comment]

We appreciate your recommendation for publication. We are grateful for your
thoughtful evaluation and constructive suggestions in this 2nd review again.
Your latest comments regarding intersegmental membrane variability and
assembly success rate analysis have provided crucial insights that helped us
refine key technical aspects of the study.

**[R2-2]** The authors mentioned that the electrode implantation sites located in
the intersegmental membrane beneath the anterior thoracic dorsal plate. Is
there any variability in the mechanical properties of the intersegmental
membrane across individuals or different life cycle stages? Are such individual
variations likely to affect the consistency of electrode implantation or the
effectiveness of stimulation?

[Reply to this comment]

Thank you for pointing out the possible impact of individual and developmental
variation. To minimize such variability, we standardized our experimental
subjects by selecting only adult male cockroaches for testing. Cockroaches
exhibit relatively consistent morphological characteristics across
developmental stages, and previous studies have reported that the mechanical
properties of the intersegmental membrane are similar among individuals, with
a Young's modulus of approximately 1 kPa²⁵. In our experiments, we did not
observe any significant differences in electrode implantation or stimulation
effectiveness across individuals among individuals of varying body sizes.

[Changes in manuscript]

The following parts were added to the Section 3.1.

*Line Number 483 – 485.*

*This study compared the performance of adult male Madagascar hissing*
*cockroaches (5 – 7 cm, 6 – 8g) with our protocol with hybrid robots*^{7,26} *with a*
*well-established stimulation protocol.*

**[R2-3]** The authors mentioned that the success rate of assembly varied among
insects with different body sizes. The 6.5-7.0 cm group only achieved the
success rate of 13%, which is much lower than the 5.5-6.0 cm group. Can the
authors discuss more on the possible reasons? Is it possible to solve it by
improving the automatic assembly strategy?

[Reply to this comment]

In our prototyping, the target body size of the cockroach is 5.0 -6.0 cm, which
represents the majority body-size range for this cockroach species. Hence, the
prototyping fits this size better than the larger individuals. It is important to figure
out the cause of failure and explore improvement especially for large-scale
production. In the failure analysis (Table. 4, Fig. S9), the most common cause
of assembly failure was hook misalignment (accounting for 85.7% of total failure
causes), indicating the mounting structure did not fit well with the larger
individuals. Hence, the primary cause of failure was the physical mismatch in
mounting structure itself rather than the automatic assembly system and its
program. To address this issue, we enlarged the hook structure by 1 mm on
each branch. This modification improved the assembly success rate to the 80.0%
for 6.0 – 7.0 mm body size group. Thank you for prompting us to elaborate on
this point. We have included the discussion in the revised manuscript.

[Changes in manuscript]

The following parts were added to the Section 2.3.4.

*Line Number 387 – 389.*

*Hence, hook failure was the main reason for the failed assembly trials (85.7%*
*of the total assembly failures for 6.0 -7.0 cm groups, Table 4).*

*Line Number 407 – 410.*

*The implementation of an enlarged mounting structure (with a 1 mm expansion*
*per hook branch) significantly improved the assembly success rate to 80.0%*
*for 6.0 - 7.0 cm cohort (Fig. S10), effectively mitigating the hook failure issues*
*previously observed in larger-size groups.*

**Reviewer #3:**

**[R3-1]** The author has addressed most of my concerns and added the
requested experiments and corresponding content. However, there are still
some minor problems as follows. I recommend acceptance after minor
revisions.

[Reply to this comment]

We are grateful for your thorough review and constructive suggestions
especially electrode specifications, backpack electronics details, and coverage
calculation methodology, which helps clarifying key technical aspects of the
study. We appreciate your time and expertise strengthening the research.
Thank you for recommending the acceptance.

**[R3-2]** Does the width of the bipolar electrode only have to be less than 1.2mm?
Does it need to be greater than a minimum?

[Reply to this comment]

The bipolar electrodes were designed to be placed on the left and right sides of
the insect to induce directional turning through unilateral stimulation. In our
experiments, an inter-electrode width of 1.2 cm was found to be effective for
both implantation and stimulation. While we ensure that the electrode width
cannot exceed this value due to anatomical constraints, a minimum width is not
systematically investigated. This is because significantly narrower electrode
spacing might result in less effective stimulation, potentially due to reduced
lateral separation or decreased stimulation selectivity. Therefore, a minimum
threshold is not considered in this study.

**[R3-3]** Please provide the specific model numbers and electrical parameters
(including operating voltage and power consumption) of the backpack
electronic components, along with battery specifications and corresponding
operational duration. It is recommended to include a schematic diagram of the
final battery configuration in the carried device.

[Reply to this comment]

Main components of the backpack microcontroller include CC1352 (1.8 – 3.8 V,
5.8 mA at Active mode Rx, 2.9 mA at Active mode MCU 48MHz), DAC AD5504
(2.3 – 5.5 V, 2 mA), and Sub-1GHz antenna (passive components, no operating

voltage or power consumption). The battery used in this study is 3.7 V, 50 mAh.
Based on the test with cockroaches for the covering mission, the battery can
stably supply power for 20 – 30 mins. The configuration of the battery in the
carried device was provided in Fig. R4.

[Changes in manuscript]

Fig. R4 was added to Fig. 3C and Fig. 5A.

*Fig. R4. Configuration of battery on the insect. A 3.7 V, 50 mAh battery was*
*sticked to the backpack using double-sided tape.*

**[R3-4]** How is methyl salicylate distributed in the arena? Is it uniformly
distributed, only at the starting point, or does it have a stepwise distribution? At
what point do you consider choosing such a distribution?

[Reply to this comment]

Methyl salicylate was not distributed in the arena but applied to the hindleg tarsi
of the cockroaches before their release to the terrain. This approach was
adopted assuming that in the real-world scenarios, distributing the target arena
might be impractical as human operators may not have access to the area.
Hence, instead of distributing methyl salicylate in the terrain, we directly applied
to the insects' sensory organ, which was proved effectiveness on boosting the
insects' motion²⁷. Sorry that the previous version of the manuscript did not
clearly explain that point. The current revised one includes clear explanation.
Thank you.

[Changes in manuscript]

The following parts were added to the Section 3.6.

*Line Number 711 – 715.*

*Hybrid robots were released from the designated area's corner. Before their*
*release, a chemical booster, methyl salicylate²⁷ was applied to the hindleg tarsi*
*of the insects. Such chemical was proven effective to improve insects' motion*
*activeness level for covering mission.*

**[R3-5]** Can you provide some specific examples, such as some motion
sequence clips from the navigation task in Fig. 5B(iii), to illustrate the role of the
deceleration command in sharp turns, dead-end paths, or highly confined
regions?

[Reply to this comment]

As shown in Fig. 5B(i), the terrain used in our navigation task was relatively
open and did not include sharp turns, dead-end paths, or highly confined
regions. Therefore, the specific effects of deceleration in such challenging
scenarios were not found in the experiment. The previous studies^{1,2} discussed
and concluded the importance of reduced locomotion speed for enhancing
obstacle negotiation. However, direct and controllable deceleration had not
been realized prior to our work. Our study successfully achieved deceleration
control through electrical stimulation, enabling more flexible and precise
regulation of the insects' movement dynamics. This capability is crucial for
improving navigation performance, especially when negotiating complex
terrains.

**[R3-6]** Please provide the specific calculation method of covered space, that is,
the radius of the covered area of each cockroach at each time point.

[Reply to this comment]

Thank you for the advice. Yes, it is better to add it. In this paper, we did not use
radius of the covered area to calculate the covered space. Instead, we equally
divided the target terrain into 20 * 20 small squares (Fig. R5A). Each small
square has edge of 10 cm. As long as one cyborg insect passed by a small
square, that square was labelled as covered (Fig. R5B, covered squares are
shaded). Otherwise, that square was labelled as uncovered. Thus, the covered
space was calculated as

$$S = n_{covered} \times S_{square}$$

where, $n_{covered}$ is the number of the covered squares and S_{square} is the area of
the small square, which is fixed as 100 cm².

We selected trajectory of insect 1 from the Fig. 5B. iii as an example to explain
 how we did the calculation. At certain time point, i.e., 10 min, the insect
 trajectory was recorded as shown in Fig. R5B. $n_{covered}$ was 40. Hence, the
 covered space for this cockroach at this time point is $40 * 100 \text{ cm}^2$, i.e., 4000
 398 cm^2 .

*Fig. R5. Calculation of the covered area. (A) The target terrain was divided into*
 *20×20 squares. (B) The squares were labelled as “covered” when the insects*
 *passed.*

[Changes in manuscript]

Fig. R5 was added to the Fig. S11.

The following part was added to the Section 3.6.

*Line Number 722 – 729.*

*The terrain was divided into 400 squares, each measuring $10 \times 10 \text{ cm}^2$ (S_{square})*
 *for easy coverage computation. Any hybrid robot passing through a particular*
 *square deemed that region covered. The number of covered squares is noted*
 *as $n_{covered}$. The covered area (S) and coverage rate (C) were calculated as*
 *below,*

$$S = n_{covered} \times S_{square} \quad (1)$$

$$C = \Delta S / \Delta t \quad (2)$$

*where $n_{covered}$ is the number of covered squares, ΔS is the change of the*
 *covered area, and Δt is the change of time (Fig. S11).*

=====
=====

Overall, we thank all the reviewers for their comments and advice. We are
especially grateful for the reviewers' time and effort in reviewing the manuscript
twice and for their constructive feedback aimed at improving the quality of our
work. Their continued engagement has greatly contributed to strengthening the
final version. We will continue our efforts to further develop this technology so
that it may contribute to saving lives in real-world scenarios.

Sincerely,

All the authors represented by

Hirotaka Sato, Professor

Nanyang Technological University, Singapore

References:

- 1. Fabian, S. T., Sumner, M. E., Wardill, T. J. & Gonzalez-Bellido, P. T.
Avoiding obstacles while intercepting a moving target: a miniature fly's
solution. *Journal of Experimental Biology* **225**, jeb243568 (2022).
- 2. Lecoœur, J., Dacke, M., Floreano, D. & Baird, E. The role of optic flow
pooling in insect flight control in cluttered environments. *Sci Rep* **9**,
7707 (2019).
- 3. Latif, T. & Bozkurt, A. Line following terrestrial insect biobots. in
*Proceedings of the Annual International Conference of the IEEE*
*Engineering in Medicine and Biology Society, EMBS* 972–975 (2012).
doi:10.1109/EMBC.2012.6346095.
- 4. Vo Doan, T. T., Tan, M. Y. W., Bui, X. H. & Sato, H. An Ultralightweight
and Living Legged Robot. *Soft Robot* **5**, 17–23 (2018).
- 5. Ariyanto, M. *et al.* Teleoperated Locomotion for Biobot between Japan
and Bangladesh. *Computation* **10**, (2022).
- 6. Tsukuda, Y. *et al.* Calmbots: Exploring possibilities of multiple insects
with on-hand devices and flexible controls as creation interfaces. in *CHI*
*Conference on Human Factors in Computing Systems Extended*
*Abstracts* 1–13 (2022).
- 7. Lin, Q. *et al.* Resilient conductive membrane synthesized by in-situ
polymerisation for wearable non-invasive electronics on moving
appendages of cyborg insect. *npj Flexible Electronics* **7**, (2023).
- 8. Latif, T., Whitmire, E., Novak, T. & Bozkurt, A. Sound Localization
Sensors for Search and Rescue Biobots. *IEEE Sens J* **16**, 3444–3453
(2016).
- 9. Tran-Ngoc, P. T. *et al.* Intelligent Insect–Computer Hybrid Robot:
Installing Innate Obstacle Negotiation and Onboard Human Detection
onto Cyborg Insect. *Advanced Intelligent Systems* **5**, (2023).
- 10. Dražanský, M., Charvát, M., Macek, I. & Mohelníková, J. Thermal
Imaging Detection System: A Case Study for Indoor Environments.
*Sensors* **23**, (2023).
- 11. Li, R., Lin, Q., Tran-Ngoc, P. T., Le, D. L. & Sato, H. Smart insect-
computer hybrid robots empowered with enhanced obstacle avoidance
capabilities using onboard monocular camera. *npj Robotics* **2**, (2024).
- 12. Ma, S. *et al.* The autonomous pipeline navigation of a cockroach bio-
robot with enhanced walking stimuli. *Cyborg and Bionic Systems* **4**,
0067 (2023).

- 13. TDK InvenSense. MPU-9250 Product Specification.
<https://invensense.tdk.com/products/motion-tracking/9-axis/mpu-9250/>
(2016).
- 14. Thanh Tran-Ngoc, P. *et al.* *Gait-Adaptive Navigation and Human*
*Searching in Field with Cyborg Insect.* (2024).
- 15. Huo, R. *et al.* The trapped human experiment. *J Breath Res* **5**, (2011).
- 16. Statheropoulos, M. *et al.* Dynamic vapor generator that simulates
transient odor emissions of victims entrapped in the voids of collapsed
buildings. *Anal Chem* **86**, 3887–3894 (2014).
- 17. Diskin, A. M., Španěl, P. & Smith, D. Time variation of ammonia,
acetone, isoprene and ethanol in breath: a quantitative SIFT-MS study
over 30 days. *Physiol Meas* **24**, 107 (2003).
- 18. Das, T. *et al.* A Mobile Robot for Hazardous Gas Sensing. in *2020*
*International Conference on Computational Performance Evaluation,*
*ComPE 2020* 62–66 (Institute of Electrical and Electronics Engineers
Inc., 2020). doi:10.1109/ComPE49325.2020.9200082.
- 19. Bílek, J., Maršolek, P., Bílek, O. & Buček, P. Field test of mini
photoionization detector-based sensors—Monitoring of volatile organic
pollutants in ambient air. *Environments* **9**, 49 (2022).
- 20. Karami, H. *et al.* Application of gas sensor technology to locate victims
in mass disasters – a review. *Natural Hazards* Preprint at
<https://doi.org/10.1007/s11069-024-06809-5> (2024).
- 21. Senseair AB. Senseair S8 CO₂ Sensor.
<https://senseair.com/product/s8/> (2023).
- 22. Sensirion AG. Sensirion SCD30 CO₂ Sensor Datasheet.
https://sensirion.com/media/documents/4EAF6AF8/61652C3C/Sensirion_CO2_Sensors_SCD30_Datasheet.pdf (2020).
- 23. DFRobot. Fermion: MEMS NH₃ Sensor.
[https://wiki.dfrobot.com/SKU_SEN0567_Fermion_MEMS_NH3_Sensor](https://wiki.dfrobot.com/SKU_SEN0567_Fermion_MEMS_NH3_Sensor_breakout)
[_breakout](https://wiki.dfrobot.com/SKU_SEN0567_Fermion_MEMS_NH3_Sensor_breakout) (2023).
- 24. Hsueh, T. J. & Ding, R. Y. A Room Temperature ZnO-NPs/MEMS
Ammonia Gas Sensor. *Nanomaterials* **12**, (2022).
- 25. Vincent, J. F. V & Wegst, U. G. K. Design and mechanical properties of
insect cuticle. *Arthropod Struct Dev* **33**, 187–199 (2004).
- 26. Erickson, J. C. *et al.* Effective stimulus parameters for directed
locomotion in Madagascar hissing cockroach biobot. *PLoS One* **10**,
(2015).

27. Lin, Q., Kai, K., Nguyen, H. D. & Sato, H. A newly developed chemical
locomotory booster for cyborg insect to sustain its activity and to
enhance covering performance. *Sens Actuators B Chem* **399**, 134774
(2024).